

# A global-scale two-layer transient groundwater model: development and application to groundwater depletion

Inge E. M. de Graaf[1], Rens (L. P. H.) van Beek[1], Tom Gleeson[2], Nils Moosdorf[3], Oliver Schmitz[1], Edwin H. Sutanudjaja[1], and Marc F. P. Bierkens[1,4]

[1]Department of Physical Geography, Faculty of Geosciences, Utrecht University, the Netherlands
[2]Civil Engineering, University of Victoria, Canada
[3]Department of Biochemistry / Geology, Leibniz center for Tropical Ecology, Germany
[4]Unit Soil and Groundwater Systems, Deltares, Utrecht, the Netherlands

*Correspondence to:* Inge de Graaf, Department Geology and Geological Engineering, Colorado School of Mines, USA, idegraaf@mines.edu

**Abstract.** Groundwater is the world's largest accessible source of freshwater to satisfy human water needs. Moreover, groundwater buffers variable precipitation rates over time, thereby effectively sustaining river flows in times of droughts as well as evaporation in areas with shallow water tables. Lateral flows between basins can be a significant part of the basins water budget, but most global-scale hydrological models do not consider surface water-groundwater interactions and do not include a lateral groundwater flow component. In this study we simulate groundwater head fluctuation and groundwater storage changes in both confined and unconfined aquifer systems using a global-scale high-resolution (5 arc-minutes) groundwater model by deriving new estimates of the distribution and thickness of confining layers. Inclusion of confined aquifer systems (estimated 6% to 20% of the total aquifer area) changes timing and amplitude of head fluctuations, as well as flow paths and groundwater-surface water interactions rates. Also, timing and magnitude of groundwater head fluctuations are better estimated when confining layers are included. Groundwater flow paths within confining layers are shorter then paths in the underlying aquifer, while flows within the confined aquifer can get disconnected from the local drainage system due to the low conductivity of the confining layer. Lateral groundwater flows between basins are significant in the model, especially for areas with (partially) confined aquifers were long flow paths are simulated crossing catchment boundaries, thereby supporting water budgets of neighboring catchments or aquifer systems. The two-layer transient groundwater model is used to identify hotspots of groundwater depletion resulting in an estimated global groundwater depletion of 6700 km$^3$ over the 1960-2010, consistent with estimates of previous studies.

## 1 Introduction

As the world's largest accessible source of freshwater, groundwater plays a vital role in satisfying the basic needs of human society (Gleeson et al., 2016). It serves as a primary source of drinking water and supplies water for agricultural and industrial activities. During periods of low or no rainfall, groundwater storage provides a natural buffer against water shortage, preserves evaporation in areas with shallow water tables, and sustains base- flows to rivers and wetlands, thereby supporting ecosystem



habitats and biodiversity (e.g. Bierkens and van den Hurk, 2007; de Graaf et al., 2013; Wada et al., 2014). Moreover, groundwater often flows across topographical and administrative boundaries at considerable rates, supporting water budgets of receiving catchments or aquifers (Schaller and Fan, 2009). However, groundwater resources are increasingly threatened by excessive abstractions, particularly in irrigated areas where abstraction rates are high and groundwater is only slowly renewed (Gleeson et al., 2011). The most direct effects of groundwater depletion are falling water tables and the irreversible loss of storage. As a consequence, pumping costs increase, and baseflows to rivers and wetlands are reduced, negatively affecting ecosystems and one of the causes to land subsidence.

Despite the importance of groundwater and the explicit treatment of groundwater recharge and pumping in global studies (Wada et al., 2010; Döll et al., 2014), most global-scale hydrological models do not include a groundwater flow component. The main reason of this omission is the lack of consistent global-scale hydrogeological information. Such data is needed to obtain a realistic physical representation of the groundwater system, allowing for a more realistic simulation of groundwater head dynamics and lateral flows (e.g Fan et al., 2007; Schaller and Fan, 2009) and are especially needed when these models move to finer resolutions (e.g. 5 arc-minutes) (Wood et al., 2012; Bierkens et al., 2015).

Previous work on global-scale groundwater models has been done by e.g. Fan et al. (2013) and de Graaf et al. (2015). The study of Fan et al. (2013) produced a first high-resolution global groundwater table map. However, their method does not include hydrogeological information on aquifers (e.g. depths and transmissivities). In addition the hydrological connection between groundwater and rivers, which is the primary drainage of groundwater in humid regions, is ignored. Moreover, their model requires calibration to head observations and only returns the steady state head distribution. The recent study of de Graaf et al. (2015) presents the first high-resolution global-scale groundwater model including hydrogeological information and accounting for groundwater- surface water interactions. Lateral groundwater flows for an upper, unconfined aquifer at the steady-state are simulated. A relative simple method for aquifer parameterization is used based on available global datasets of lithology (Hartmann and Moosdorf, 2012) and permeability (Gleeson et al., 2011) such that the method provides results for data-poor environments. Also, aquifer thickness is derived globally using terrain attributes. The results are promising, shown by the high correlation between observed and simulated averaged groundwater heads. However, the study's greatest limitation is that only the steady-state head distribution is reported and temporal changes in groundwater head patterns due to climate or human impacts are not considered. Also, only a single unconfined aquifer is described and vital information on the accessibility and quality of the groundwater water resource is missing. This information is needed to simulate the aquifer sensitivity to storage changes due to climate fluctuations or abstractions. Also, abstractions are preferentially positioned in confined and partially-confined aquifer systems, as these systems are less sensitive for climate fluctuations and groundwater quality is often better than from unconfined systems (Foster and Chilton, 2003).

The first objective of this study is to represent the groundwater system more realistically, including aquifers with a confining layer, and simulate groundwater head dynamics affected by changes in climate and human water use. The groundwater system is more realistically described by including information on the presence of confining layers, leading to the categorization of world's aquifers in (partially) confined and unconfined systems. The improved physical groundwater representation is expected to lead to a more realistic estimate of aquifer sensitivity to changes in storage by climate or human impacts.





The second objective of this study is to provide estimates of groundwater depletion and head declines worldwide. Previous estimates of groundwater depletion (e.g. Döll et al., 2012; Wada et al., 2010; Konikow, 2011; Pokhrel et al., 2012) vary by an order of magnitude. For example the flux-based approaches of Wada et al. (2010) estimate groundwater depletion at 283 ± 40 km$^3$yr$^{-1}$ for 2010 as the residual of grid-based groundwater recharge and withdrawal. Their approach overestimates

depletion as it does not account for increased capture due to decreased groundwater discharge, nor for long-distance surface water transports. The volume-based approach of Konikow (2011) estimates groundwater depletion at 145 ± 39 km$^3$yr$^{-1}$ between 2001 and 2008 based on data for the US and other big aquifer systems by extrapolation of groundwater depletion globally using the average ratio of depletion to abstractions observed in the US. Our study will be the first to simulate changes in global groundwater storage in relation to climate and groundwater pumping while including lateral groundwater flow and

groundwater-surface water interaction. We expect our depletion estimate to be lower than earlier flux-based estimates and more similar to volume-based estimates.

## 2 Methods

### 2.1 General

In this study a two-layer groundwater model for the terrestrial part of the world was developed (excluding Greenland and

15 Antartica). The model consists of two parts: (1) the hydrological model PCR-GLOBWB (van Beek et al., 2011) and (2) the groundwater model using MODFLOW (McDonald and Harbaugh, 2000). Both models were run at 5 arc-minute resolution (approx. 10 km$^2$ at the equator) over the period 1960-2010. The groundwater model was forced with outputs from PCR-GLOBWB, specifically via groundwater recharge and river discharges (Fig. 1). A brief description of the models and coupling is given here, a more detailed description is given in de Graaf et al. (2015).

### 2.1.1 Global hydrological model

The model PCR-GLOBWB simulates hydrological processes in and between two vertically stacked soil stores (maximum thickness 0.3 and 1.2 m respectively) and one underlying groundwater store. The model was run at 5 arc-minutes resolution at a daily time step. For the climate forcing, monthly data from CRU TS 2.1 (Mitchell and Jones, 2005) was downscaled using ERA-40 (Uppala et al., 2005) and ERA-interim (Dee et al., 2011) reanalysis to obtain a daily climate forcing (see de Graaf

et al. (2013) for a more detailed description of this forcing dataset). For each time step and every grid cell the water balance of the soil column is calculated based on the climatic forcing that imposes precipitation (rain or snow depending on temperature), reference potential evapotranspiration, and temperature. Vertical exchange between the soil and groundwater occurs through percolation and capillary rise. In the original version of the PCR-GLOBWB model no lateral groundwater flow is simulated. Instead, groundwater dynamics is described by means of a linear storage-outflow relation (Store 3 in Fig. 1A) . Specific surface

runoff snowmelt, interflow, and baseflow are accumulated along the drainage network that consists of laterally connected





surface water elements representing river channels, lakes, or reservoirs. A kinematic wave approximation at a sub-daily time step is used to route surface water to the basin outlet.

PCR-GLOBWB also simulates human water use by calculating at each time step sectoral water demand (irrigation, industrial, domestic, livestock), the associated water withdrawals (partitioned into groundwater and surface water based on avail-
ability and pumping capacity), water consumption and return flows. Data on human water demand and water withdrawal were adopted following the methods described in de Graaf et al. (2013) and Wada et al. (2014). Sectoral water demands were estimated using country statistics and population numbers downscaled to 5 arc-minutes resolution ($\sim$10 km$^2$). To approximate economic development over the model period, data of Gross Domestic Product, electricity, and household consumption were used. Industrial water demands were kept constant over the year, while domestic demand reflects seasonality in relation to air temperature. Irrigation demand was calculated assuming optimal crop growth, accounting for bare soil evaporation and soil water availability (see Wada et al. (2014) for more details on the calculation of irrigation water demand). Total water demand was dynamically attributed to surface water and groundwater withdrawals, based on the simulated availability of each water resource, and return flows to groundwater and surface water simulated (see de Graaf et al. (2013) for more details on the dynamic attribution of water demand and the estimation of return flows).

### 2.1.2 Global groundwater model

A two-layer MODFLOW model (McDonald and Harbaugh, 2000; Schmitz et al., 2009) replaces the linear groundwater store of PCR-GLOBWB. Aquifer properties were prescribed and include (partially) confined and unconfined aquifer systems (discussed in the next paragraph). The MODFLOW model was forced with outputs from PCR-GLOBWB, specifically the flow between layer 2 and 3 (Fig. 1) consisting of net recharge (recharge minus capillary rise) and river discharge. The net recharge is the input for the MODFLOW's recharge (RCH) package, while the MODFLOW's river (RIV) and drain (DRN) packages are used to incorporate interactions between the groundwater and surface water. As PCR-GLOBWB runs on a Gaussian grid (lat-lon projection) we have to take account of the fact that cell areas and volumes of the MODFLOW grid do not vary in space. This is done by adjusting recharge and the storage coefficient in the RCH and BCF packages to account for varying cell areas and volumes associated with a lat-lon grid (see Sutanudjaja et al. (2011) and de Graaf et al. (2015) for details). Next to the river levels, the boundary conditions of the global groundwater model were obtained as fixed heads set equal to global sea-level (reference level 0). Initial conditions were obtained by a steady state-state simulation of hydraulic head with average recharge and then warming up the model by running the year 1960 back-to-back for 10 years to reach dynamic equilibrium.

Three levels of groundwater-surface water interactions are distinguished: (1) large rivers, wider than 25 m, (2) smaller rivers with a width smaller than 25 m, and (3) springs and streams higher up in the valley.

1. For large rivers, interactions are governed by actual groundwater heads and river levels ($HRIV$ [m]). The latter was calculated from river discharges ($Qchn$ [m$^3$s$^{-1}$]) simulated by PCR-GLOBWB and using channel properties based (channel width and depth) based on geomorphological relations to bankfull discharge ($Qbkfl$ [m$^3$s$^{-1}$]) (Lacey, 1930; Savenije, 2003; Manning, 1891). The RIV-package calculates the water flux between the river and groundwater $Qriv$





[m$^3$d$^{-1}$]. Water infiltrates the aquifer if the groundwater head lies below the river head: $Qriv$ is then positive. Water is drained from the aquifer when groundwater head lies above the river head: $Qriv$ is then negative. $Qriv$ for the larger rivers ($Qriv.Big$ [m$^3$d$^{-1}$]) was calculated as:

$$Q_{\text{riv.Big}} = \begin{cases} c \times (\text{HRIV} - h) & \text{if } h > \text{RBOT} \\ c \times (\text{HRIV} - RBOT) & \text{if } h \leq \text{RBOT} \end{cases} \tag{1}$$

where $C$ [m$^2$d$^{-1}$] is the river bed conductance calculated from river length, river bed width and depth and river bed resistance (assumed 1 day), $h$ [m] is groundwater head, and $RBOT$ [m] is the river bottom calculated for bankfull conditions (taken as a rule of thumb happening every 1.5 year). Surface elevation ($DEM$ [m]) is taken as the reference level here.

2. Smaller rivers are defined as drains; water can only leave the groundwater system through the drain when heads get above
the drainage level. In this case, the drainage levels were taken equal to the $DEM$. The drainage was then calculated as $Qriv.Small$ [m$^3$d$^{-1}$]:

$$Q_{\text{riv.Small}} = \begin{cases} c \times (\text{DEM} - h) & \text{if } h > \text{DEM} \\ 0 & \text{if } h \leq \text{DEM} \end{cases} \tag{2}$$

3. $Qriv.Big$ and $Qriv.Small$ quantify flow between streams and aquifers and are the main components of the baseflow, $Qbf$, which is negative when water is drained from the aquifer. At the 5 arc-minute resolution however, the mean
stream is insufficient to represent local sags, springs, and streams higher up in the mountainous area. We assume that groundwater above the floodplain level can be tapped by local springs that are represented by means of a linear storage-outflow relationship. To be consistent with the RIV and DRN packages, this linear storage term is also negative when water is drained. Total drainage $Qbf$ [m$^3$d$^{-1}$] is thus calculated as:

$$Q_{bf} = (Q_{riv.Big} + Q_{riv.Small}) - (J \cdot S_{3,flp}) \tag{3}$$

where $S_{3,flp}$ [m$^3$] is the groundwater storage above the floodplain (obtained from PCR-GLOBWB) and $J$ [d$^{-1}$] is the recession coefficient parameterized based on Kraaijenhof van der Leur (1958):

$$J = \frac{\pi T}{4S_y L^2} \tag{4}$$

where $T$ [m$^2$d$-1$] is transmissivity, $Sy$ [-] is the specific yield (for each hydrolithological unit; Tabel 1) and $L$ [m] is the average distance between streams and rivers as obtained from the drainage density (van Beek et al., 2011).





## 2.2 Summary of model coupling

PCR-GLOBWB and MODFLOW were only coupled one-way. PCR-GLOBWB is first run over the period 1960-2010 with daily time steps. Next, the following outputs were passed to MODFLOW as monthly averages: surface water levels are passed to the RIV package, net recharge (groundwater recharge minus capillary rise) is passed to the RCH package and groundwater

abstractions are passed to the WELL package. Finally, MODFLOW was run over the period 1960-2010 at monthly time steps with these fluxes and boundary conditions imposed.

This type of coupling ensures that the same amount of water passes through the groundwater model as through the groundwater zone (Fig. 1A store 3) of PCR-GLOBWB and also yields the same global average flux between surface and groundwater. Thus the coupling ensures full terrestrial balance closure. Also, it implicitly includes capillary rise, which is calculated in

PCR-GLOBWB (van Beek et al., 2011) and is included in the net recharge. However it does not include feedback effects of groundwater levels on the partioning of surface fluxes as a full coupling would do.

## 2.3 Aquifer parameterization

The aquifer parameterization is based on available global datasets on lithology (Global Lithology Map (GLiM), Hartmann and Moosdorf (2012)) and permeability (Gleeson et al., 2011, 2014), combined with an estimate of aquifer thicknesses (de Graaf

et al., 2015). A detailed description of the aquifer parameterization is given in (de Graaf et al., 2015), a summary is given below since the focus of this study is to extend the aquifer parameterization by categorizing world' s aquifers in confined and unconfined systems.

### 2.3.1 General: Aquifer permeability and transmissivity

Aquifer permeabilities were defined by lumping the lithology classes defined by Hartmann and Moosdorf (2012) to 7 combined

hydrolithologies (adopted from Gleeson et al. (2011)), representing broad lithological categories with similar hydrogeological characteristics. These units were subdivided further on the basis of texture for unconsolidated and sedimentary rocks (Table 1), resulting in a global map representing regional scale permeability. The polygons of the lithological map, delineating a hydrogeological unit, were subsequently gridded to 30 arc-seconds (approx. 1 km$^2$ at the equator) and aggregated as the geometric mean at 5 arc-minutes resolution (approx. 10 km$^2$ at the equator). To calculate transmissivities ($T$ [m$^2$d$^{-1}$]) aquifer

thicknesses are needed. These have been estimated since no global dataset of observed aquifer thickness is available. Using the assumption that productive aquifers coincide with sedimentary basins and sediments below river valleys, the distinction is made between (1) mountain ranges, and (2) sediment basins representing the aquifers. Subsequently, aquifer thicknesses were estimated by relating these to terrain attributes (e.g. curvature) after calibration of these relations with reported aquifer thicknesses from U.S. groundwater modeling studies (see de Graaf et al. (2015) for an extensive description of this procedure

to delineate aquifer systems and estimate thicknesses).





### 2.3.2 Delineation of confining layers

In this study we delineate confining layers and categorize the aquifers of the world in (partially) confined and unconfined aquifers. The categorization is done using information on grain sizes of unconsolidated sediments in the lithological map (GLiM) and additional information in the sediment property description of GLiM. Figure 2 shows the resulting map of con-
fining layers. Some clear spatial inconsistencies can be seen, e.g. for the US over the High Plain aquifer, or the German-Polish border (inconsistencies are discussed in Hartmann and Moosdorf (2012)). We interpreted the GLiM lithologies of 'fine-grained unconsolidated sediments' and 'mixed-grained unconsolidated sediments' as potential near-surface confining units. About 30% of the mixed grained unconsolidated sediments have layers of clay, silt or till in the sediments description. We assume, based on the uncertainty in the sediment descriptions, that this 30% represents the minimum percentage of unconsolidated mixed
grained sediments with a confining unit. The maximal percentage is if all mixed grained unconsolidated sediments would have a confining unit. These two percentages are used to test the sensitivity of groundwater patterns and fluctuations to absence or presence of confining layers.

Additionally, we classified regions of the world where we know confining layers are present but are not fully represented in GLiM; the coastal zones. We reconstructed the inland extent of Holocene fine-grained coastal plains that overlay older,
coarser sediments from the Pleistocene. A reconstruction of sea-level rise (e.g. Toscano and Macintyre (2003)) and coastal sedimentation (e.g Syvitski et al. (2005)) implies that 6000 years ago along many Holocene coastal plains the coastline was more seawards compared to the modern situation. Coastlines have receded tens of kilometers since in many deltas (e.g. Stanley and Warne (1994)). With sea-level rise during the Holocene, the downstream gradient along the river flattened, and in the downstream part sediments accumulated. The apex of the delta and deposition areas indicate the point where accumulation
originated. This accumulation point is reconstructed using the profile curvature along the river, with its occurrence where the curvature changes from convex to concave starting from the coast. The accumulation points were identified globally for larger rivers (>25 m wide) with their outlet on, or near the coast (i.e. within 50 km). Points at high elevations, clearly not related to sedimentation in the coastal zone, were excluded.

Figure 3 (top) shows a schematic cross-section of a coastal aquifer with a confining layer on top. Thickness of the con-
fining layer at the coast is estimated using ocean bathymetry (elevation taken one 5 arc-minute grid-cell out of the coast). The thickness at the coastline is interpolated along the river using the river gradient until the apex, where thickness is minimal. The interpolation is done for all identified large rivers and thickness of the coastal plains is derived by interpolation between the rivers, bounded by the elevation contours on which the apex are situated (Fig. 3 bottom). Following this approach, approximately 11% of the global coastline is classified as confined after interpolation.

### 2.3.3 (Partially) confined aquifers: aquifer transmissivity and storativity

Confined aquifers are defined here as those aquifers overlain by a fine grained confining layers. We realize that the term 'confined' aquifers may not entirely cover the different hydrogeological settings that are found in reality. First, aquifers covered with fine-grained sediments may still allow vertical flow through the confining layers and are thus in fact leaky or semi-confined



aquifers. Second, due to the varying thickness and permeability of the confining layers certain areas may be in fact unconfined, depending on the spatial scale under consideration. Third, in areas with excessive groundwater pumping the drawdown close to wells may cause part of the originally (pressurized) groundwater head to fall below the top of the aquifer, causing it to be partially unconfined. All these cases mean that what we define here as confined aquifers may be better termed as partially
and semi-confined aquifers. Having noted this limitation, for the sake of briefness of terminology, we elect to call all aquifers covered with fine grained confining layers 'confined aquifers' ,including the hydrogeological settings described above. Finally, we note that in the implementation in MODFLOW we use a global two-layer model where for the areas with unconfined aquifers the top layer is given the same hydraulic properties as the underlying aquifer.

The parameter settings used for the unconsolidated confined aquifers and the confining layers are shown in Fig. 4. For all
other rocks and sediments the values from Table 1 are used. In absence of any other information, vertical conductivity was initially assumed to be the same as the horizontal conductivity, but a calibration procedure is used to calibrate the storage coefficient, the horizontal conductivity as well as the anisotropy ratio kh/kv assuming kh $\geq$ kh. Aquifer storage capacities were parameterized using specific yields for unconfined aquifers (Table 1) and for confined aquifers a storage coefficient of 0.001 (Heath, 1983) was assumed. Thickness of coastal confining layers is estimated using ocean bathymetry and interpolation (as
described in 2.3.2). For 80% of the coastal confined gridcells the estimated confining layer thickness is approximately 10% of the total estimated aquifer thickness. Based on this finding and by lack of any other information, we decided to apply a thickness of 10% of the total estimated thickness to all confining layers not located near the coast.

## 2.4   Analyzing the effect of schematization

We analyse the effects of incorporating confined aquifer systems on simulated groundwater head fluctuations, groundwater flow
patterns, groundwater-surface water interactions and storage changes. Three scenarios were formulated describing different spatial distributions of confined and unconfined aquifer systems: (1) unconfined systems only (following de Graaf et al. (2015)), (2) confined systems for fine grained unconsolidated sediments and mixed grained unconsolidated sediments with fine grained layers (30%) (in GLiM) and coastal confined regions (minimum confining scenario), and (3) confined systems for fine grained and all mixed grained unconsolidated sediments and coastal confined regions (maximum confining scenario). The parameter
settings of confined and unconfined systems are used as described above (Fig. 4 and Table 1).

## 2.5   Calibration and validation of simulated groundwater heads

In order to calibrate and validate the groundwater model, time-series were selected from the US (available from the USGS: www.usgs.gov/water) and Europe (Rhine-Meuse delta: Sutanudjaja et al. (2011)). The used time-series have a record covering at least five years and include seasonal variation; for the US ~28000 stations, for Europe ~6000 stations were available. In
case multiple stations were positioned in one grid cell, we chose not to average these but use them as they are, because the different stations generally have observations over different time periods. Also, stations have different surface elevations or are placed in different aquifers (i.e. deep or shallow). We randomly selected half of the observations for calibration and half for validation (split sample approach).





Limited calibration was used to better fit the groundwater model to the head observations. The calculation times (approximately 2 weeks for a 1960-2010 run) were too long to allow for an full-fledged automated calibration (e.g. Doherty (2015)). As a first step, we ran the groundwater model in steady state to calibrate the horizontal conductivity and the anisotropy ratio of the confining layers. A single global pre-factor was used varying the horizontal and vertical conductivities between 0.01 and

100 times the initial value, changing the anisotropy ratio $kh/kv$ (a total of 9x9 = 81 runs). Using the parameters of the best performing run (the parameter setting with the minimum root mean square error RMSE) we then further calibrated the model performing six transient runs while changing the storage coefficient between 0.1 and 5 times the initial value and allowing the horizontal conductivity and anisotropy ratio to be changed +/- 10%. This calibration was performed using the maximum confining scenario, as we see this as the most realistic scenario. The best parameter set (with minimum RMSE) found was also

used in the minimum confining scenario.

The model performance was validated by evaluating simulated- to observed water table head time-series (the other half of the head observation locations not used for calibration). Heads (with reference to ocean level) instead of depths were used as heads measure potential energy and are therefore more meaningful than depths. The evaluation focused on mean monthly values, timing of fluctuations (given by the cross-correlation coefficient $R$), and amplitudes. The latter is compared by the

(relative) inter-quantile range error, $QRE$, calculated as:

$$Q_{RE} = \frac{IQ_{7525}^m - IQ_{7525}^o}{IQ_{7525}^o} \qquad (5)$$

where, $IQ_{7525}^m$ and $IQ_{7525}^o$ are the inter-quantile ranges of the modeled and observed data time series.

The cross-correlation is calculated as:

$$R = \frac{s_{mo}}{s_m s_o} \qquad (6)$$

where $s_m$ and $s_o$ are the sample standard deviations of the modelled (m) and observed (o) samples, and $s_{mo}$ is the sample covariance between the two. Results are given in the form of histograms.

Apart from these global validation measures we also evaluated the performance of the model in three areas where groundwater depletion is severe (as follows from the global analysis, sections 2.7 and 3.5): The High Plains aquifer (USA), The Central Calley of California (USA) and India. For these areas, we compared on an aquifer-by-aquifer basis the average head decline

rates (shown in the supplementary online information).

## 2.6 Groundwater flow patterns

The effect of considering confined aquifers on groundwater flow patterns is analyzed by simulating groundwater flow paths. Groundwater flow paths and travel times were simulated for heads averaged over the simulation period, comparing two model setups: unconfined and confined aquifer systems (the unconfined and maximum confining scenarios). Flow paths were cal-

culated with particle tracking in MODPATH (Pollock, 1994), using cell- to-cell flux densities [volume/time/area] as input. In





MODPATH a flow path is computed by tracking the particle through the subsoil from one cell to the next, from the location of infiltration toward the point of drainage.

Following Schaller and Fan (2009) the spatial redistribution of local recharge was evaluated using the ratio between groundwater drainage and recharge: $Qgw/Rgw$. The ratio was estimated for sub-basins that we defined using stream-orders, starting from the second stream-order at 5 arc-minutes. The local drainage network was defined at the 5 arc-minute grid-cell, and for every cell a stream is present. It follows that, for a ratio of 1, all water recharged in the basin is also drained in the basin, or cross basin- boundary groundwater inflows and outflows are approximately equal. If part of the water recharged in a catchment flows to a neighboring catchment and it is not compensated by cross-boundary inflow, the catchment's groundwater drainage is less than groundwater recharge, thus $Qgw/Rgw< 1$. The catchment is then classified as a 'net groundwater exporter' . If more water is drained than recharged in a catchment, the catchment is classified as a 'net groundwater importer' and $Qgw/Rgw>1$. Deviations from 1 are primarily a function of geology and topography, while climate and basin size influence the magnitude of these deviations (Schaller and Fan, 2009). The spatial pattern of net importers and exporters is expected to change by including confined aquifer systems.

## 2.7 Analyzing the global effects of human groundwater use

We used the two-layer transient global groundwater model to analyze the effects of human groundwater withdrawals on groundwater heads. The groundwater abstractions taken from the 1960-2010 run with PCR-GLOBWB (including human water use) were subsequently used as input for the MODFLOW model through the WELL-package. For confined systems all abstractions were located in the confined aquifer, for unconfined systems abstractions were located in the lower layer, which has the same conductivity and storage coefficient as the top layer. The MODFLOW model was additionally forced with PCR-GLOBWB river discharge and net recharge outputs (recharge minus capillary rise) that include the effects of abstractions and corresponding return flows over the model period 1960-2010. We subsequently analyzed the head change over the period 1960-2010 from the simulations with the global groundwater model. The PCR-GLOBWB runs show that over the past decades, total water demand increased globally from approximately 900 $km^3y^{-1}$ for 1960 to 2000 $km^3y^{-1}$ for 2010 (Wada et al., 2010). Total groundwater abstractions increased globally from approximately 460 $km^3y^{-1}$ for 1960 to 980 $km^3y^{-1}$ for 2000 (de Graaf et al., 2013). These values are in the same range as previously reported rates by the International Groundwater Assessment Centre: 321 $km^3y^{-1}$ for 1960 to 734 $km^3y^{-1}$ for 2000 (www.un-igrac.org).

## 3 Results

## 3.1 Delineation of confining layers

Figure 2 shows the spatial distribution of the confining layers. For the minimum confining scenario about 6% of the total aquifer area is classified as confined, i.e. belonging to either coastal confined, or fine grained and layered mixed grained unconsolidated sediments (in GLiM (Hartmann and Moosdorf, 2012)). This 6% is assumed to be the minimum extent of



confined aquifers systems worldwide. For the maximum confining scenario about 20% of the total aquifer area is classified as confined, belonging to either coastal confined or fine grained and mixed grained unconsolidated sediments. The relative distribution per continent does not differ much between the two scenarios (bar plots of Fig. 2). Combining the two scenarios 5.39 x $10^6$ km$^2$ to 17.34 x $10^6$ km$^2$ of the world's aquifers are classified as confined.

## 3.2 Calibration and validation

Table 4 shows the parameter ranges of horizontal and vertical hydraulic conductivities and storage coefficients used in the calibration and the resulting 'best' parameter set (with minimum RMSE). With these parameters the model performance was evaluated for the three scenarios (unconfined, minimal, or maximal confining) focusing on mean monthly values, timing of fluctuations and amplitudes of groundwater heads. Simulated head fluctuations were compared to the observed head time series (for Europe and US) not used in the calibration. Note that the locations of these head time series are often biased towards river valleys, coastal ribbons, and productive aquifers. Examples of time-series of the observed and simulated heads for the three scenarios are shown in Fig. 5. Instead of plotting actual head values, we plotted the anomalies related to their mean values. For these examples we can conclude that the model is able to capture both timing (good timing when $R > 0.5$) and amplitude (small error when $|Qre| < 50\%$) of the observed heads reasonably well. Also, the amplitude is better captured when a confined aquifer system is included, in particular in the Central Valley example.

The histograms of grid-cell maximum R (eq. 6), and minimum $|Qre|$ (eq. 5) for the three scenarios are given in Fig. 6 and show overall good agreement in timing ($R > 0.5$), which slightly improves when confined aquifer systems are included, showing larger frequencies in classes 0.8-0.9 and 0.9-1 Also amplitude errors are small ($|Qre| < 50\%$), and cover not only shallow groundwater depths but deeper groundwater depths as well (Fig. 6). When including confining layers, the amplitude errors increase slightly shifting from 0-25% to 25- 50% and from 25-50% to 50-75%. Figure 6 shows that model results are promising and shows that we are able to mimic groundwater head fluctuations well, in particular when confined layers are considered. The scatter and statistics (Fig. 7) of temporal mean simulated values against mean observed values for the maximum confining scenario show that averaged values are reproduced well (very high $R^2$ and $\alpha$ close to 1) and are slightly better when only head locations in areas with sediments and sedimentary basins are considered in the comparison. Model performance of the other scenarios is comparable.

## 3.3 Global map of groundwater depth

Figure 8 (top) shows the long-term average groundwater depth simulated for the maximum confining scenario simulated for a situation without groundwater pumping. General patterns in groundwater depths can be identified, and are similar for the three scenarios and comparable to previous results by de Graaf et al. (2015). At the global-scale, sea level is the main control of groundwater depth and throughout the entire coastal ribbon shallow groundwater tables are found. These areas expand inland where the coastal ribbon or coastal plains meets the major river deltas, like the Mississippi, Indus, and large wetland areas of e.g. South America. At the regional scale, recharge is the main control together with topography. Topography influences for example the heads in the flat areas of central Amazon and the lowlands of South America which receive groundwater from



the higher elevated areas. High, flat recharge regions also result in shallow groundwater tables, e.g. the tropical swamps of the Amazon. Regions with low recharge rates show deep groundwater tables; the deserts stand out where groundwater, if present, is now disconnected from the local topography. Also, for mountain regions deep groundwater tables are simulated. In these areas local aquifers in sedimentary pockets in mountain valleys are smaller than the grid resolution ( < 10 km ) and are therefore not
captured. As a result, groundwater heads in these regions are likely underestimated (de Graaf et al., 2015).

The differences in groundwater depth between the two model layers (top: confining layer, bottom: confined aquifer) are small. Insets of parts of the US and Europe illustrate this (Fig. 8 bottom). For unconfined systems, where top and bottom layers have equal conductivities, any groundwater depth differences are expected to be small. For confined systems, where conductivities are different for the top confining layer and bottom confined aquifer, the groundwater heads in the confined
aquifers are often higher than those in the overlying layers. This is for example seen along the Dutch coast, Po river, Rhone, and along the Gulf of Mexico, including the Mississippi. However, when groundwater recharge does not seep through the confining layer groundwater tables can exceed the hydraulic heads of the confined aquifer. This is seen for example in the Mississippi embayment and the Garonne region (France).

### 3.4 Groundwater flow patterns

Flow paths simulations were performed using averaged groundwater heads (1960-2010) for the unconfined and maximum confining scenarios. We focus the discussion on part of the US (Fig. 9) as this region contains shallow and deep groundwater tables and confined and unconfined aquifer systems, and has been the focus of related previous studies (Schaller and Fan, 2009; Gleeson et al., 2011). Confined systems are e.g. found for the High Plain aquifer, along the Gulf of Mexico, and Mississippi embayment (Fig. 9 left). The simulated flow paths show how groundwater recharge is redistributed over time and space (Fig. 9
middle and right). Longer flow paths, crossing catchments boundaries, provide additional recharge to the importing catchment, thereby supporting water budgets in that catchment. The difference between the unconfined and confining scenario (maximum) is evident (Fig. 9). Travel distances and travel times are generally shorter if confining layers are present. This results from the lower permeability of the upper layer, providing higher groundwater tables, higher drainage densities and as a consequence more local groundwater systems. Occasionally flow paths end up in the lower aquifer where they become disconnected from
the surface water system over longer distances resulting in longer travel times. The flow lines in the case of unconfined aquifers are much longer with longer travel times as groundwater tables are generally deeper and groundwater systems are larger in size.

The spatial distribution is also shown by the ratio between groundwater baseflow and groundwater recharge ($Qbf/Rch$) per a sub-basin (Fig. 10). In this study the sub-basin is defined by stream-order, starting from the second stream-order. A ratio
of 0.1 means 90% of the recharged water is exported, a ratio of 2 or larger means at least half of the groundwater drainage comes from neighboring catchments. The histograms in Fig. 10 show the effect of confining units on the distribution of net groundwater importer and exporter sub-basins. The results show that the change in distribution of groundwater exporter and importer sub-basins is small when confining layers are introduced: there is a small shift from sub-basins.





### 3.5 Analyzing the global effects of human groundwater use

Figure 11 shows, for the maximum confining scenario, a global map of the total volume (per grid cell) of groundwater removed from storage by pumping. The well-known hotspots of groundwater depletion appear, e.g. High Plains aquifer, California, Indus basin, and Saudi-Arabia (e.g. de Graaf et al. (2013); Richey et al. (2015); Scanlon et al. (2012). As an additional means

of validation we compared the simulated head drops by groundwater depletion with reported rates in a number of well-known stressed aquifers (Gleeson et al., 2012; Richey et al., 2015): the Central Valley, High plains aquifer, and India (see Fig. 13 in the Supplementary Online Information). These results show that trends in California, the Southern Great Plains and India are reasonable to good (of similar order of magnitude), while those in the northern part of the Great Plains are severely over-estimated. Possible explanations for this mismatch are an underestimation of aquifer permeability and storage coefficient,

overestimation of groundwater abstraction rates and the underestimation of enhanced surface water infiltration.

We compared the simulated depletion trends for the maximum confining scenario and unconfined scenario, calculated using head declines and storage coefficients (Fig. 12). We also compared these estimates to the difference between recharge and groundwater abstraction (in case of deficit) per grid-cell, which is the way that depletion rates were estimated using flux-based methods (Wada et al., 2010; Pokhrel et al., 2012). The total trend for the maximum confining scenario shows a globally

close to zero depletion between 1960 and 1980, after which a steady increase is observed until 1998 after which we see an accelerated depletion until 2005 and a steady rate thereafter. The lack of global depletion over 1960-1980 must be the result of temporally increased capture from surface water, as we observe an almost linear increase in abstraction minus recharge over the same period. The sudden increase in depletion after 1998 is most likely an interplay between climate (e.g. the 1997-1998 strong El Nino) and increased surface water and groundwater withdrawals related to socio-economic trends (Wada et al., 2010).

Figure 14 (Online Supplementary Information) shows that effect of hydraulic properties on the groundwater depletion volumes (note volumes, not heads) is considerable, which makes estimates of groundwater depletion by volume-based methods rather uncertain.

Using the maximum confining scenario and the best parameter set, the total groundwater depletion over 1960-2010 is estimated as 6290 km$^3$ and over 1960-2000 at 3200 km$^3$ which is comparable to previous published estimates of volume and

flux-based approaches (see Table 3). Figure 12 reveals that the unconfined scenario shows in general the same trend. However the estimated groundwater depletion is larger. The estimated depletion difference is 5210 km$^3$. This difference can be explained by the increase in river capture under the confined scenario. The presence of a confining layer results in a larger area of influence of head decline.

Despite the lower permeability of the confining layer, this larger area of influence causes a larger decrease in baseflow or a

larger increase in riverbed filtration. Indeed, the groundwater drainage rate of the confined aquifer model is 50 km$^3$y$^{-1}$ lower than that of the unconfined aquifer model, which adds to 3000 km$^3$ over 1960-2010.

The estimated depletion calculated as the deficit between groundwater recharge and abstraction for cells where more water is abstracted than recharged (flux-based method as used in e.g. Wada et al. (2010); Pokhrel et al. (2012)) shows much bigger depletion volumes of 18960 km$^3$. Again, the difference in estimated depletion can be explained by the increase in river capture,





which is not accounted for by calculating the recharge-abstraction difference but is included in the groundwater model. The large difference confirms the need to use a lateral groundwater model, accounting for groundwater- surface water interactions, when sensitivity of aquifers to storage changes is studied.

## 4   Conclusions and discussion

This paper presents a global-scale groundwater model simulating lateral flows and head fluctuations over the period 1960-2010. It is the first global-scale two-layer transient groundwater model that includes both unconfined and confined aquifer systems, and groundwater-surface water interactions. The model's aquifer parameterization is based on global datasets of surface geology and hydraulic properties and topography-based estimates of the vertical structure of the aquifer systems. Only globally available data sets are used in order to keep the methods readily applicable to data-poor environments. The world's

aquifers are classified into confined and unconfined systems to understand aquifer sensitivity to groundwater abstractions and to properly project future groundwater level declines.

Inclusion of confined aquifer systems (estimated 6% to 20% of the total aquifer area) changes timing and amplitude of head fluctuations, as well as flow paths and groundwater- surface water interactions rates. Model performance on average heads and timing of fluctuations is slightly better when confined systems are considered. Groundwater flow paths within confining

layers are shorter compared to paths in the aquifer, while flows within the confined aquifer can get disconnected from the local drainage system due to the low conductivity of the confining layer. This change in groundwater hydrology is reflected in head fluctuations and flow magnitudes.

Lateral groundwater flows between basins are significant in the model, especially for confined aquifer areas where long, slow paths are simulated crossing catchment boundaries, thereby supporting water budgets of neighboring catchments or aquifer

systems.

The estimated global depletion over the period 1960-2010 is 6292 km$^3$. This value is in the range of earlier published values (e.g. 8000 km$^3$ Wada et al. (2010), 5000 km$^3$ Wada et al. (2012)). Hotspot regions of depletion are found for intensively irrigated areas, like the High Plains aquifer (630 km$^3$ for 2000) or Central Valley (30 km$^3$ for 2000). Although depletion is strongly overestimated for the Northern High Plains compared to reported data ($\sim$350 km$^3$ reported by USGS (Scanlon et al.,

2012)), simulated temporal fluctuations mimic measured data well. The same holds for Central Valley, where we underestimate total depletion ( which is $\sim$60 km$^3$). The comparison of depletion volumes for confined, unconfined, and estimated as the deficit between recharge and abstraction, showe increased river capture lowers the estimated depletion volume. This confirms that a lateral groundwater model, with realistic parameter settings for its aquifers, is preferred when sensitivity of aquifers to groundwater abstractions is studied.

Of course there are several limitations to our approach. Firstly, the grid-resolution (5 arc-minutes, approx. 10 km at the equator) is still too coarse to capture local aquifers in higher and steeper terrain. Groundwater heads are underestimated for these areas. Note however, that perched water tables within the soil are accounted for in the PCR-GLOBWB.





Secondly, our groundwater model is not fully coupled to our hydrological model, meaning the groundwater-surface water interaction is still simplified: two-way groundwater-surface water exchange and the resulting groundwater recharge is first simulated in PCR-GLOBWB. This and the resulting surface water levels are imposed on the MODFLOW RIV package afterwards. Although this approach can be expected to preserve the same groundwater-surface water fluxes in the groundwater model, the

effects of groundwater pumping on the surface water system are more intricate and would preferably require a tighter coupling. In future work the two models will be fully coupled to incorporate feedback effects on a daily timestep basis.

Thirdly, the global hydrogeological model, although at the edge of what can be expected when using only global information, leaves much room for improvement. A concerted effort of the hydrogeological community to use information from regional on hydrological- and groundwater models, as well as observations, to update the global two-layer model would likely lead to the

largest improvement in global groundwater assessments (Bierkens et al., 2015).

This modelling study is the next step towards a fully-coupled global surface water- groundwater model. Although further improvements are possible and needed, our model already proved to be a useful and relevant tool to simulate effects of climate variability and water abstractions on groundwater head variability and head decline. Knowledge on this is vital and needed to ensure sustainable water use, particularly for the semi-arid regions of the world, where groundwater abstractions will intensify

due to an increase in drought frequency and duration combined with population growth, expanding irrigation areas, and rising standards of living.

*Acknowledgements.* This study was funded by the Netherlands Organization for Scientific Research (NWO) under the program 'Planetary boundaries of the global fresh water cycle'. This work was carried out on the Dutch national supercomputer Cartesius with the support of SURFsara.



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




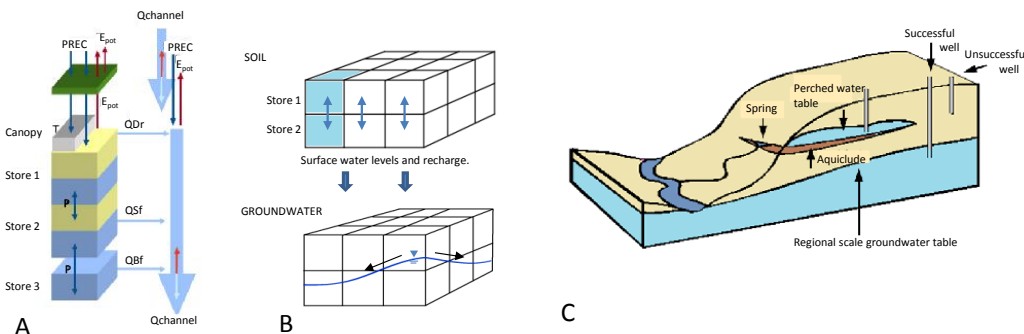

**Figure 1.** A) Original model structure of the hydrological model PCR-GLOBWB. B) The groundwater store (store 3 in PCR-GLOBWB) is replaced by a groundwater model. The groundwater model is forced with net groundwater recharge minus capillary rise) and surface water levels calculated with PCR-GLOBWB. C) A cross-section illustrating the difference between the simulated regional-scale groundwater levels and perched water levels. The latter are simulated as interflow in PCR-GLOBWB.

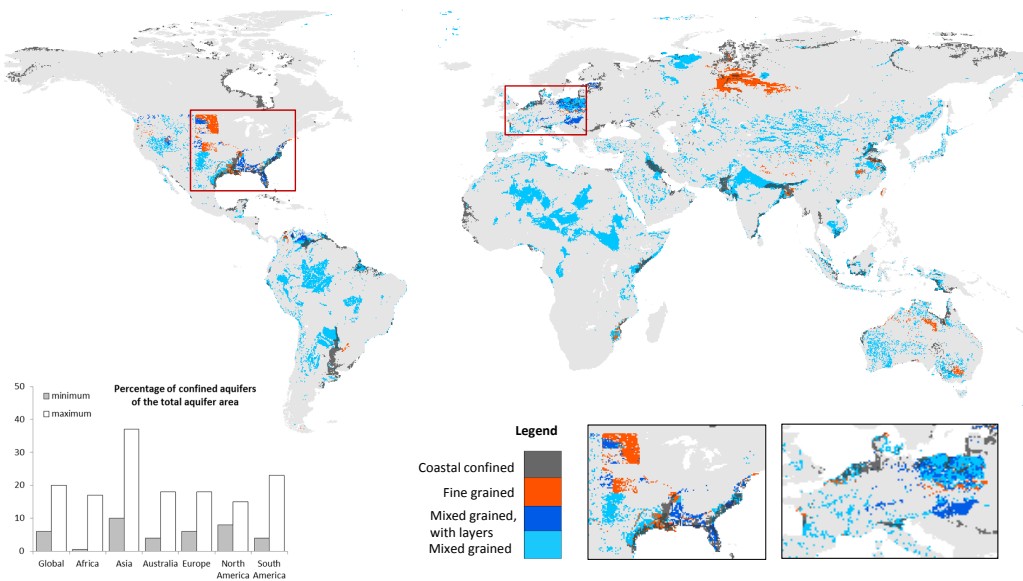

**Figure 2.** Map of defined confining layers for unconsolidated sediments. The insets illustrate the difference in quality of the input data used for the lithological map in more detail. The histogram shows the global percentage of confining layers assumed for the minimum and maximum scenario, and per continent.



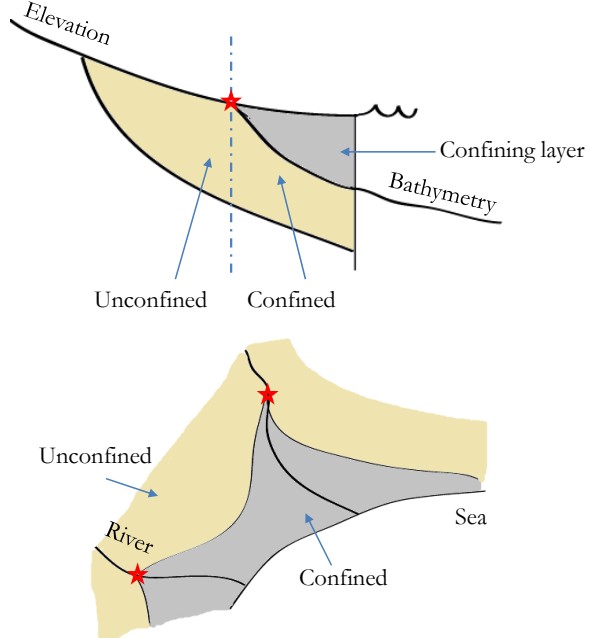

**Figure 3.** Top) 2-dimensional schematization of coastal confined aquifer classification. The star indicates the apex. Bottom) spatial extent coastal confined aquifer.

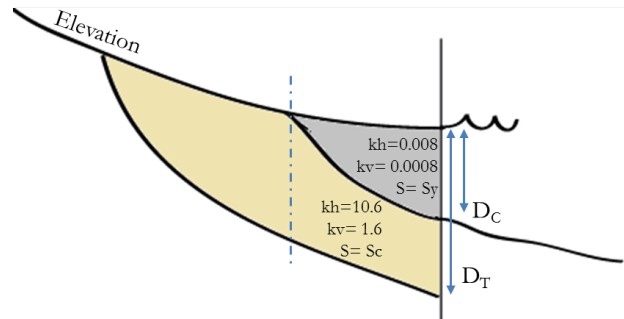

**Figure 4.** Used aquifer parameter settings for unconsolidated sediments for confined aquifer systems. $D$ is thickness [m], $D_T$ is the total estimated aquifer thickness, $D_c$ is estimated thickness of the confining layer. For the confining layer horizontal and vertical conductivities ($kh$ and $kv$ [md$^{-1}$]) were taken similar to fine grained unconsolidated sediments (see Table 1), for the confined aquifer $kh$ and $kv$ were taken similar to coarse grained unconsolidated sediments. $S$ is the storage coefficient, defined as specific yield (confining layer, values Table 1) or storage coefficient (confined aquifer).



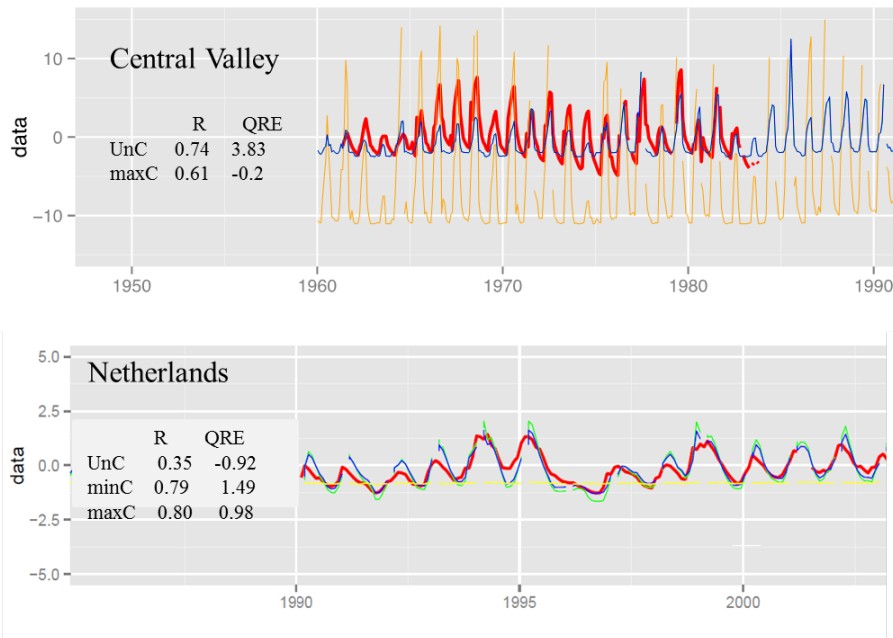

**Figure 5.** Examples of groundwater head anomaly comparison between measured data (red) and simulated data from the different scenarios; unconfined (yellow), minimal confined (green), maximal confined (blue) for a location in the Central Valley of California (USA) and a location in the Netherlands. We selected these two locations to show the difference between an unconfined (Central Valley) and a confined (Netherlands) system, for two larger aquifer systems





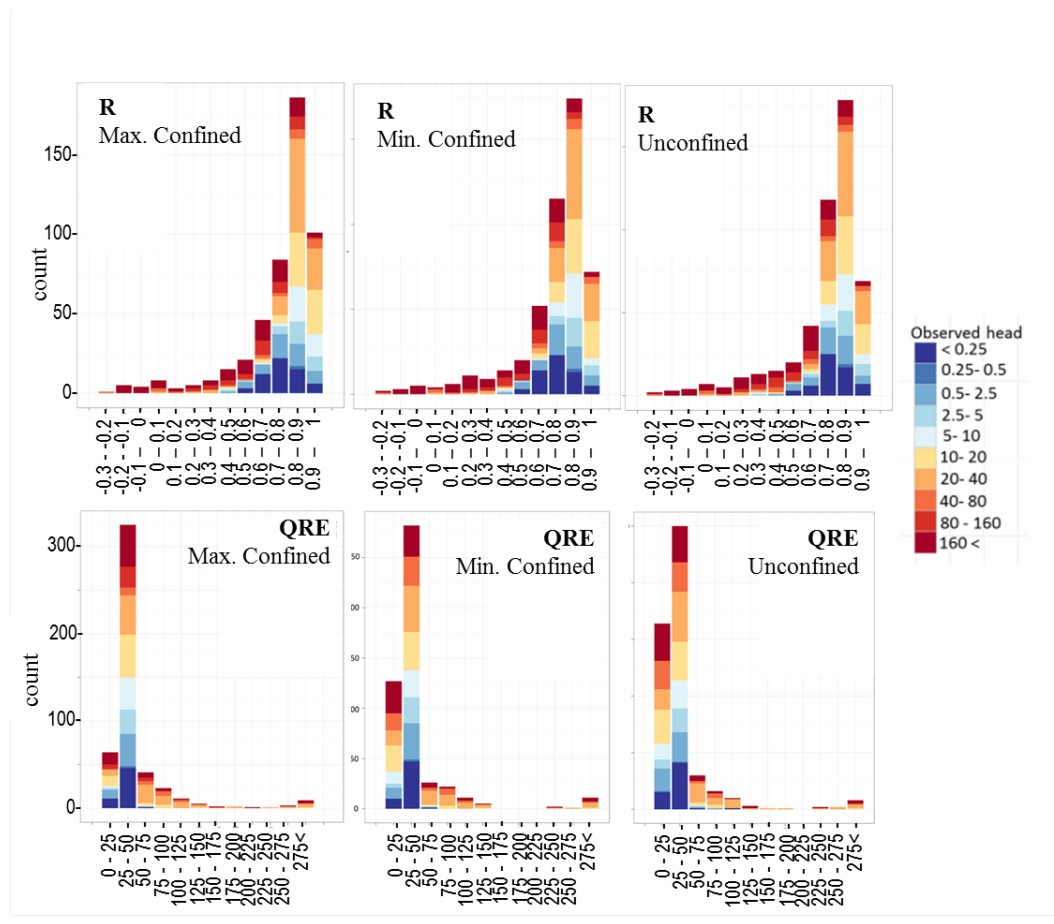

**Figure 6.** Histograms of correlation ($R$) and amplitude error ($|QRE|$) for the different scenarios. Each bar in the histogram shows clustered observed groundwater depth values, averaged over the model period (1960-2010). When more than one observed head time-series is available in a grid-cell, the result shows the highest model performance value.





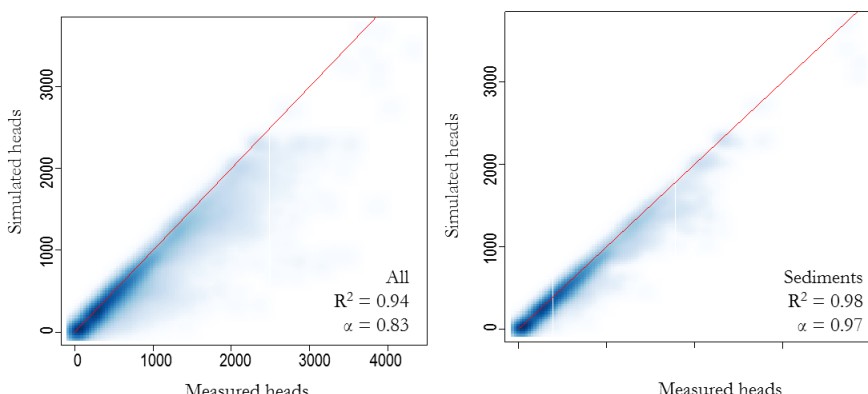

**Figure 7.** Comparing averaged observed heads against simulated heads for the maximum confining scenario, for mountain ranges (in general deep groundwater levels) and basins (in general shallow groundwater tables). $R^2$ is coefficient of determination and $\alpha$ the regression coefficient of the line passing through the origin.




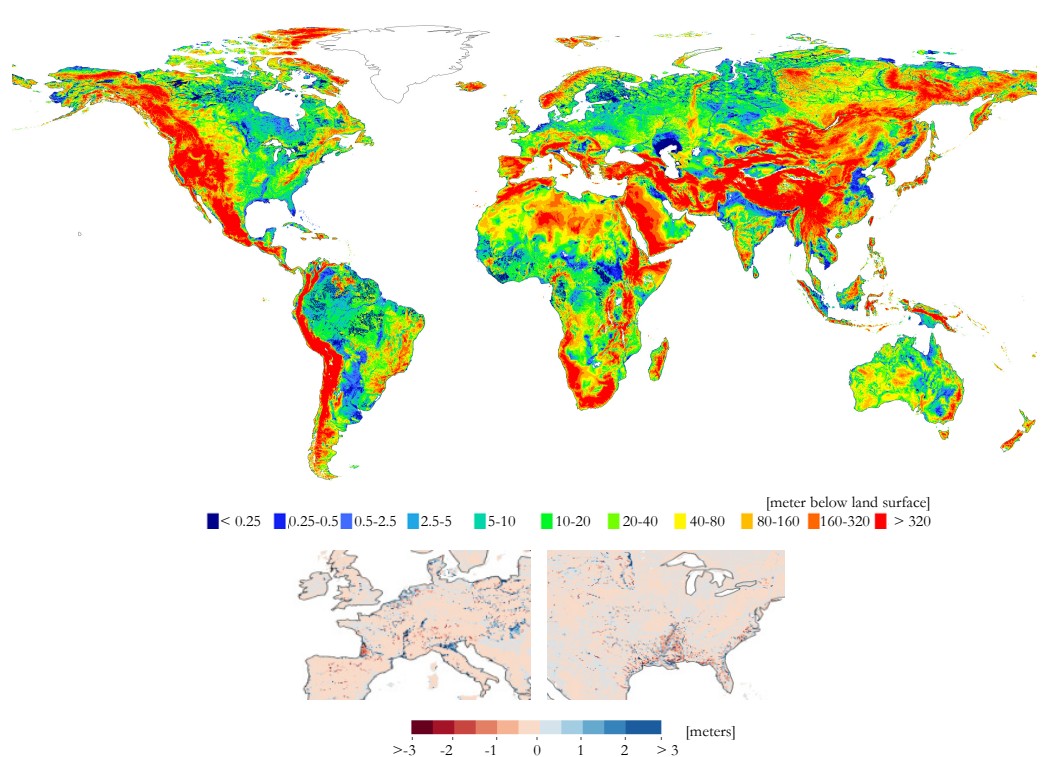

**Figure 8.** Top) Simulated average water table depths (averaged for the period 1960-2010) in meters below the land surface in the upper aquifer under natural conditions (without human water withdrawal) for the maximum confining scenario and the best parameter set. Bottom) head differences; bottom minus top layer. For water under pressure in the confined aquifer (bottom layer) a positive value is shown. When heads in the confining layer are higher, a negative values are shown.



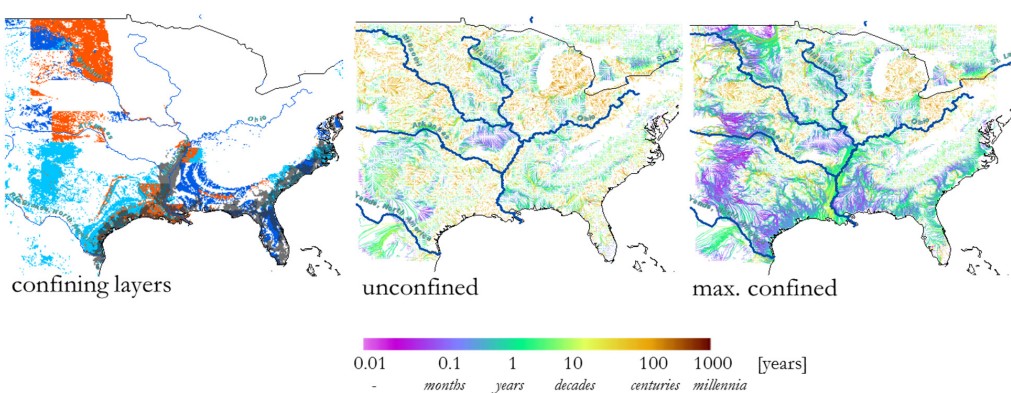

**Figure 9.** Defined confining layers and simulated flow paths for a part of the US for the unconfined and maximum confined scenario. Flow paths are overlain by major rivers.

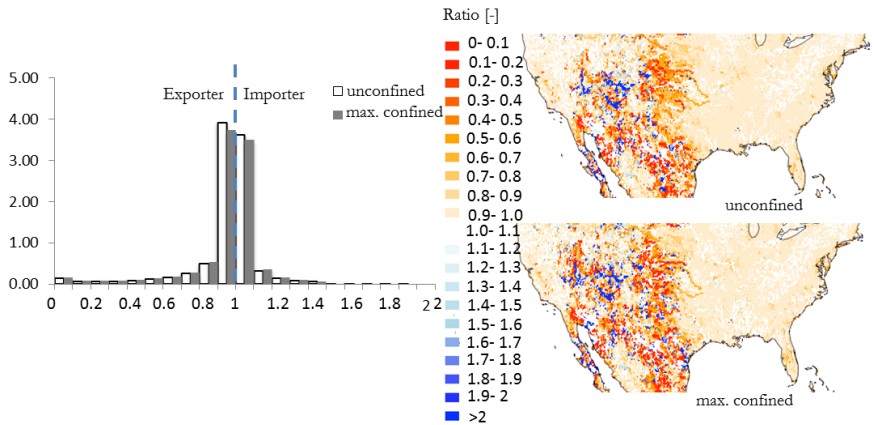

**Figure 10.** The ratio of groundwater baseflow to recharge (both in mm year$^{-1}$) distinguishing groundwater importing ($Qbf/Rch>1$) and exporting ($Qbf/Rch<1$) sub-basins. Left: frequency of sub-basins found at a given $Qbf/Rch$ ratio: comparing the results of maximal confining and unconfined scenarios. Right: Zooms for $Qbf/Rch$ ratio, shows the spatial distribution. White areas present missing values where recharge is 0.





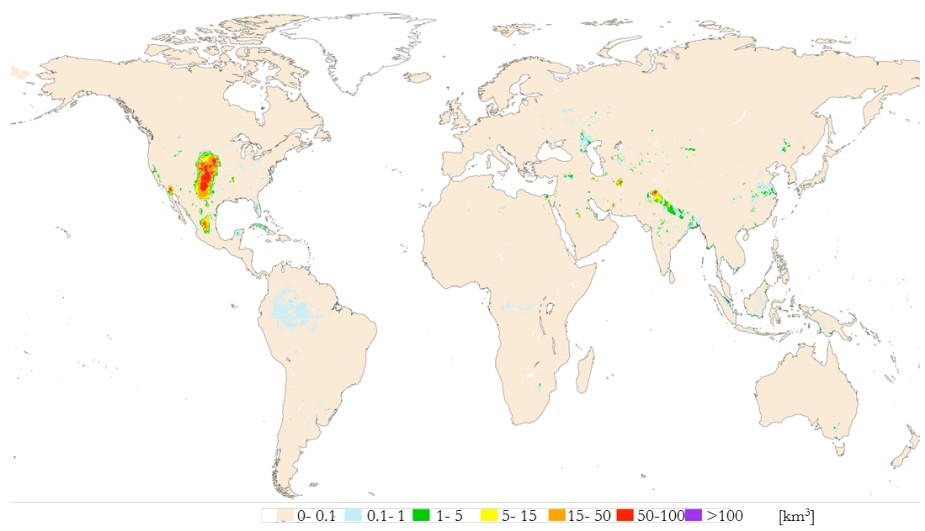

**Figure 11.** Cumulative groundwater depletion volume (over 1960-2010) resulting from human water use for the maximum confining scenario and the best parameter set.

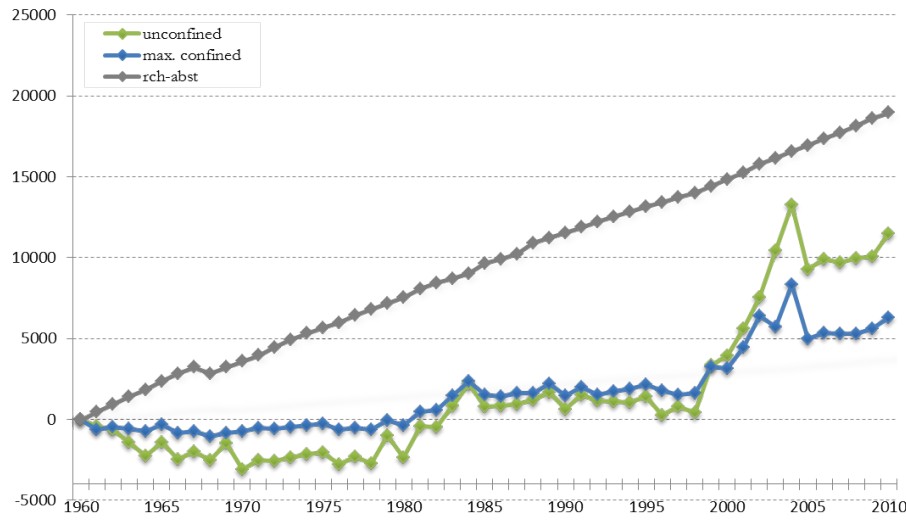

**Figure 12.** Trend of depletion globally, estimated for the maximum confining and unconfined scenario, and calculated as the cumulative deficit between recharge and abstraction for grid- cells where abstraction is larger than recharge.





**Table 1.** Lithologic and hydrolihologic categories.

| Lithologic categories[a] | Hydrolithologic categories[b] | $\log k\ \mu_{\mathrm{geo}}$ [m$^2$][b] | $\sigma$ [m$^2$][b] | $S_y$ [m/m][c] |
|---|---|---|---|---|
| Unconsolidated sediments | unconsolidated (mixed or unmixed) | −13.0 | 2.0 | 0.235 |
| | c.g. unconsolidated | −10.9 | 1.2 | 0.360 |
| | f.g. unconsolidated | −14.0 | 1.8 | 0.110 |
| Siliciclastic sediments | siliciclastic sedimentary | −15.2 | 2.5 | 0.055 |
| | c.g. siliciclastic sedimentary | −12.5 | 0.9 | 0.100 |
| | f.g. siliciclastic sedimentary | −16.5 | 1.7 | 0.010 |
| Mixed sedimentary rocks | Carbonate | −11.8 | 1.5 | 0.140 |
| Carbonate sedimentary rocks | | | | |
| Evaporites | | | | |
| Acid volcanic rocks | Crystalline | −14.1 | 1.5 | 0.010 |
| Intermediate volcanic rocks | | | | |
| Basic volcanic rocks | | | | |
| Acid plutonic rocks | Volcanic | −12.5 | 1.8 | 0.050 |
| Intermediate plutonic rocks | | | | |
| Basic plutonc rocks | | | | |
| pyroclastics | | | | |
| metamorphic | | | | |
| water bodies | not assigned | – | – | – |
| Ice and Glaciers | | | | |

[a] Hartmann and Moosdorf (2012).

[b] Based on Gleeson et al. (2011), $\log k\ \mu_{\mathrm{geo}}$ is the geometric mean logarithmic permeability; $\sigma$ is the standard deviation; f.g. and c.g. are fine-grained and coarse-grained, respectively.

[c] $S_y$ is the storage coefficient, average per category.

**Table 2.** Parameter values used in the calibration process

| Prefactors | Parameter Values | Number of Discrete Values | 'best' value |
|---|---|---|---|
| $f_{kh} \in \{10^{-2}, ..., 10^2\}$ | $kh = kh_{ori} \cdot f_{kh}$ | 9 | $kh_{ori} \cdot 1$ |
| $f_{kv} \in \{10^{-2}, ..., 10^2\}$ | $kv = kv_{ori} \cdot f_{kv}$ | 9 | $kv_{ori} \cdot 0.1$ |
| $f_{Sc} \in \{0.1, 1, ..., 4, 5\}$ | $Sc = Sc_{ori} \cdot f_{Sc}$ | 6 | $Sc_{ori} \cdot 3$ |

The subscribt 'ori' refers to the original values used in the model, as presented in Table 1





**Table 3.** Previous published depletion cumulatives over 1960-2000 (as most studies did not simulate untill 2010) compared to this study.

| Global total | km$^3$ |
| --- | --- |
| Pokhrel et al. (2012) | 18960 |
| Wada et al. (2010) | 8000 |
| Wada et al. (2012) | 5000 |
| Konikow (2011) | 2000 |
| this study | 3200 |





**Figure 13.** A) Simulated groundwater level changes over 1960 to 2010 for the High Plain and Central Valley aquifers, and simulated average groundwater level changes (over 1960-2010) for India. B) previous published data. For the High Plains measured groundwater level changes from predevelopment to 2007 are shown. For Central Valley simulated groundwater level changes from predevelopment to 1961 are shown (used fron Scanlon et al. (2012). C) Simulated depletion trends, summed over the High Plain and Central Valley. Especially depletion over the Northern High Plains are overestimated, resulting in an overestimation of total aquifer depletion compared to previous published data. Simulated values of Central Valley are compared to a previous published estimate and are in the same range.





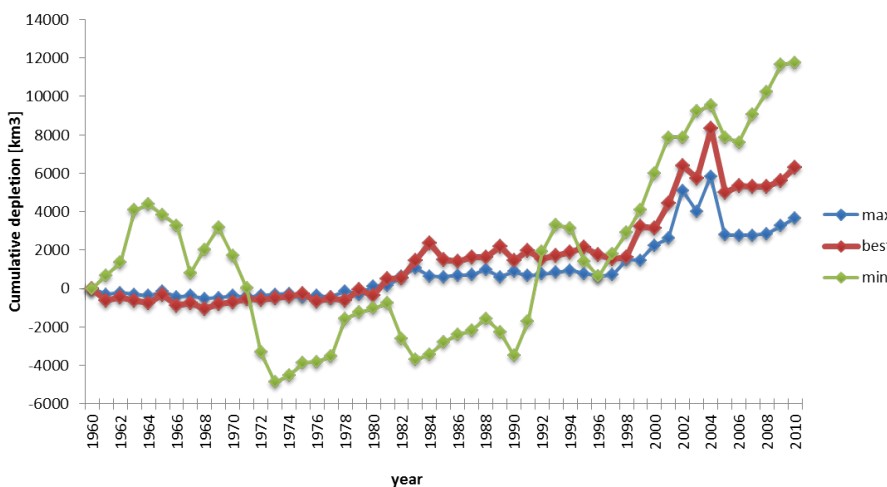

**Figure 14.** Estimated depletion using minimal and maximal parameter sets around the best peforming paramenter sets. Parameters that were changed are horizonal and vertical conductivities and storarativity.