# Peer review of "A global-scale two-layer transient groundwater model: development and application to groundwater depletion"

_Hydrology and Earth System Sciences, 2016_

## Referee Comment (RC1) · Anonymous Referee #1 · 18 Apr 2016

This paper presents an ambitious attempt at hydrologic modeling at the global scale. The work builds on a previous model of the author by adding confined aquifer units and using a transient model. Given the scarcity of physical subsurface data available at the global scale, large assumptions were made about aquifer structure and parameters. The work is clearly a step in the right direction, and we need to test our ability to model these systems, but the usefulness of the results is not clear. I have two specific concerns:

1) To calibrate a global model with observations only from the United States and from one delta in Europe doesn't seem reasonable. Especially, given that a major value of the model lies in it's ability to parameterize subsurface systems or predict groundwater

level changes for the remainder of the world, where we happen to have few observations. At this point, perhaps the model should just be applied to the US and part of Europe, where the model structure can be better tested?

2) I'm concerned about the overall discrepancy between the representative model depth, and the system it aims to simulate. This stems from using surface geology to infer aquifer properties, and similarly, using surface geology to infer the presence and properties of confining units. Many primary aquifer systems are multi-layered with numerous confining units and aquifers with varying properties with depth. The objective of the paper could be to just model the near-subsurface system. However including groundwater extraction values, which in many cases are drawn from deep systems, may force the calibration process beyond reasonable limits.

Overall, it's clear that a significant amount of work went into this, and it moves us closer to having a global groundwater model. Addressing some of the comments here about model structure, calibration, and uncertainty in storage change will help clarify the value of the model and it's results.

Specific comments:

1) It would be helpful to conceptually explain the model and assumptions a bit more clearly in the methods section. Obviously the data required to model these deep aquifers is rare, certainly at the global scale – so the current project is making reasonable assumptions in order to move the understanding forward. Given that, it should be clear early in the paper what system and dynamics it expects to model reasonably well, given the data input restrictions. Broadly, the model improves upon a previous version which modeled all aquifers as unconfined. Is the current model explicitly modeling the most surficial aquifer and most surficial confining units only? The permeability values represent the surface geology, and the confining unit permeability also seems to be based only on the shallowest layer of material.

2) Most of the large-scale (irrigation, industrial, municipal, etc.) groundwater usage is

drawn from deep wells, whose regional aquifer characteristics may not be well represented in this model. Can the authors discuss how calibrating the model with relatively shallow aquifer input parameters to fit potentially deep system extraction rates may impact the model performance? It seems like there may be a discrepancy between the system modeled and the one it is calibrated to.

3) Given a two layer model, are interactions with a shallow unconfined aquifer (e.g. alluvial aquifer overlying a confining unit) lost? Are there specific areas where surface water - groundwater dynamics were not well represented, perhaps useful for guiding future research to improve our subsurface parameterization capacity in these areas?

4) Were groundwater observations from all well depths used to calibrate the model? For the confined aquifer areas, it is highly possible that groundwater observations are being made in multiple aquifers, where deeper layers would not be expected to have a direct connection with the surface as is being modeled. I understand we cannot expect this level of detail to be included in the model, I'm just curious how fitting a model to these data will impact your results.

5) The brief description of how aquifer thicknesses were calculated (in addition to the citation to the 2015 paper) is helpful. Can a similar one be provided for how thickness of the confining unit were calculated?

6) Were the parameters for the confining units assumed based on the surface unit texture? Were any measurements (or regional model parameters) used to inform individual aquifer confining unit permeability, or were they set uniformly across the globe?

7) In the methods section 2.1.2, Does "Next to the river levels" mean proximally adjacent to? Or "next" figuratively? It sound like there are fixed head boundaries being specified at sea level adjacent to all the rivers. If this is correct, can you justify why you chose to do this? Can you explain this decision with respect to Figure 8? The depth to groundwater appears to follow topography (as you say in the paper), and is simulated quite a bit deeper than observed (e.g. much of western US and Mexico).

8) If it took 10 years for the model to reach equilibrium, does that say something about the degree of disequilibrium in the groundwater system in 1960? Do you think 10 years is reasonable? If so, or not, can you infer something about how the model is functioning?

9) There are two periods of rapid groundwater depletion in Figure 12 early 1980s and 2000s. You explain the first as being delayed despite overall abstraction > recharge (is that right?) by stream capture. Is this a process that would be included in the model, without having feedback from groundwater level on surface water?

10) The total groundwater depletion is given with 4 significant figures. Can you justify this precision? Can you provide an estimate of uncertainty on the depletion estimate based on errors associated with the groundwater level simulations and storage values?

11) The conclusion that model performance is only slightly better with the inclusion of the confined systems suggest that we do not need to model the confined systems? or that they should be modeled another way?

12) Many of the figures can be tightened up: they could use subfigure labels (A,B,C,etc.), and make sure the axis labels are final (some say "data", several are missing), and that for figures with subplots that the axis line up for all figs.

There are a handful of typos, but those can be corrected with minor effort.

---

## Short Comment (SC1) · 21 Apr 2016

Gradient-based groundwater modeling at the global scale is a big challenge, and it was very interesting to me to see results of such an effort in the manuscript of de Graaf et al. I would like to make some comments and pose questions regarding 1) the groundwater modeling and its presentation in general (as there are a few aspect that need clarification) and 2) the estimation of groundwater depletion.

1 Global groundwater modeling in general

Page 5, line 20: Regarding discharge of groundwater to local springs etc. From the wording, it is not clear if water storage S3 is from PCR-GLOBWB or the elevation of

the floodplain. Please clarify. How is the elevation of the floodplain determined, and where is it as compared to DEM, HRIV and RBOT? Please indicate in the manuscript how large this local drainage component is compared to total Qbf (globally).

Page 8, Line 4: If groundwater head falls below the bottom of the confining layer, is then the confined aquifer modelled as unconfined, and the storage coefficient set to the specific yield to Table 1 instead of 0.001?

Fig. 5: Typo in figure caption: Central Valley is unconfined, the Netherland confined.

Fig. 6: I was wondering whether all three graphs showing R and all three graphs showing QRE really have the same y-axis (they should!). The units for QRE are %?

Regarding QRE, model performance decreases according to Fig. 6, while the two examples in Fig. 5 show clearly the advantage of distinguishing between confined and unconfined aquifers, for the two wells shown. Why is this so? Can you localize where, in the confined parts of your model, assuming unconfined conditions leads to a better modeling of head amplitudes? I think it is necessary to include, in the conclusions (p. 14, l 12-14), that including confining layers leads to a worse simulation of head variations.

Fig. 9: In my understanding, Fig. 9 does not show flow paths but travel times. What exactly is shown? The travel time of the groundwater recharged at the grid cell shown? Please also clarify in the text on p. 12. I also think that your conclusion (page 14, lines 18-20) is not backed by Figs. 9 or 10. Fig. 9 shows shorter travel times in case of confined aquifer modeling, while Fig. 10 shows significant importing/exporting in mountainous areas (not flat confined areas). (Units are missing on y-axis of Fig. 10).

2 Simulation of groundwater depletion

Page 13, lines 20-25: Please define "volume-based" and "flux-based" approaches. Do you compute groundwater depletion by subtracting heads and multiplying with the storage coefficient? Do you call your approach "volume-based? Otherwise, I would not

agree with the sentence: "Figure 14 (Online Supplementary Information) shows that effect of hydraulic properties on the groundwater depletion volumes (note volumes, not heads) is considerable, which makes estimates of groundwater depletion by volume-based methods rather uncertain."

The development of the global sum of groundwater storage loss in cells in which net recharge is smaller than groundwater abstractions (as shown in Figs. 12 and 14) shows how sensitive this estimate is to parameter settings. If depletion assuming unconfined conditions only is (in 2005-2010 compared to 1960) twice the amount of that for confined conditions, I would think the model is overly sensitive. Equally important, Fig. 12, with an increase in groundwater storage before 1980 as compared to 1960, and an actual onset of groundwater depletion only in 1998 indicates to me that what you see in the first decades may be caused by the fact that 10 years of running 1960 (climate and water abstractions) on steady-state groundwater levels was not enough to get a reasonable situation of the state of groundwater heads in depletion areas in 1960. Or that the location of the groundwater table to which interaction with surface water is very sensitive was not close enough to reality (see Fig. 7 where e.g. differences of 20 m to observations are the rule but that would already have a strong impact on gw-sw interactions). How did the flows between rivers and groundwater develop over time in the depletion areas? Regarding the temporal dynamics, your simulation results show a large depletion of 5000 km3 in only 6 years (1998-2004), with relatively little dynamic at other times. Is this due to spatial averaging?

In page 4, line 27 you state that after 10 years of running 1960, a dynamic equilibrium was reached? Do you mean that after 10 years, groundwater heads did not change by +- x%? What value did you choose for x? As a test for sensitivity of Fig. 12, I suggest you rerun your model with 100 years of 1960 initialization instead of 10 years, and show the results in Fig. 14.

Another question regarding Fig. 14: The grey recharge-abstraction curve is more or less a straight line, e.g. the annual difference between groundwater recharge and

abstraction is constant between 1960 and 2010. Can you explain why, as groundwater withdrawals are known to have increased significantly during these fifty years?

In Fig. 13 C, please add observations for High Plains Aquifer, while in Fig. 13A I suggest you use the same legend/color as in 13B, for better comparability.

In Fig. 11, use mm instead of km3.

In your comparison to other estimates of global groundwater depletion, please add a comparison to the maps and global values of

Döll, P., Müller Schmied H., Schuh, C., Portmann F. T., Eicker A. (2014): Global-scale assessment of groundwater depletion and related groundwater abstractions: Combining hydrological modeling with information from well observations and GRACE satellites, Water Resour. Res., 50 (7), 5698-5720, doi:10.1002/2014WR015595.

In this study, we took into account that baseflow is reduced (and then zero) in areas of groundwater depletion (but inflow of river water into groundwater is not simulated). Also, there is groundwater recharge from surface water bodies in dry areas (very rough estimate). In addition we assumed, based on comparing our modeling results to many independent estimates including GRACE, that 70% deficit irrigation is done in groundwater depletion areas. This resulted in a best estimate of global groundwater depletion of 2240 km3 for 1960-2000, and of 3257 km3 for 1960-2009. So according to your study, gw depletion increases much faster after 2000 than in our study, and the total value is higher.

---

## Referee Comment (RC2) · Anonymous Referee #2 · 22 Apr 2016

Page 1. Line 1. This sentence appears to be the justification for building a global groundwater model, but this sentence is misleading. If by basins they mean surface-water basins, then lateral flow is only significant in the most surficial part of the system where >90% of the flow occurs. Most substantial groundwater withdrawals are from deeper confined and semi-confined aquifers where the total natural flow in the system is very small compared to the recharge, for example. It is true that most hydrological models do not include a groundwater flow component, but that in itself does not justify building a global groundwater model. Basically as the saying goes, "all models are wrong, but some are more useful than others"–so the authors have to demonstrate that their model is both accurate "enough" and "useful" to some degree. Unfortunately

[Figure]

I don't believe this model is either. It is only in the ballpark where the data exist in the 10% of the world where most of the pumping is occurring (it does a very bad job in much of the world in simply fitting depth to the water table), and it really isn't useful in the 10% of the world where the pumping is occurring because global groundwater depletion has already been quantified to a similar degree of accuracy (Wada et al, 2010, 2012; Konikow, 2011), and the groundwater model they are constructing really doesn't improve on those estimates. Lately it seems to be that the bigger the model is, the better it must be and the more likely we are to learn something new from it. Therefore the authors are now producing a global model; but in this case the model does not produce any useful results. More on this later. Page 1, line 19. This has been known for decades-why cite a reference from 2016? Why not a UN report from the 20th century? Page 2, line 29. Here the problem begins. Indeed most abstractions occur in confined aquifers (which is why the investigators include a second layer to represent them), but simulating head declines there does NOT improve any estimates of groundwater depletion. Most all groundwater depletion in deep aquifers comes out of storage, not from bounding surface-water bodies, and therefore the model -simulated depletions only mimic the abstractions that are put into the model a priori. Thus the model is only as good as the estimate of water abstractions that are available. The earlier global depletion studies did their best to estimate these abstractions. Putting these numbers into a groundwater model does not improve on them. Figure 5 horizontal axes need labels (year). "Data" is not an acceptable y-axis label—aren't these heads, in meters? Figure 6 is very hard to understand because the x-axes are not labeled and the y-axes are underlabeled—count of what? Figure 7. The points are too faint—it's not clear how far the simulated heads actually fall below the measured heads. Axis labels need to include "in meters". Figure 8. These results truly show how bad this model is. Only in the deserts of the world is the depth to the water table typically greater than 100 m. Yet most of the mountain ranges here show depths greater than 300 m. There are streams in most of those mountains so we know the depth to water is zero at those locations and likely a few tens of meters at the watershed divides. If so much of the

world gives results this bad it does not give me confidence in the rest of the world, and there really isn't any reason to do a model of the whole world, especially when 90% of the abstraction occurs in 10% of the area. The authors will argue that these areas don't have any significant depletion anyway, but that argues that those areas really don't need to be modeled. If the time and resources were focused on the 10% of the world where depletion was really occurring, you would get better results and might learn something. Figure 11. This figure makes no sense. If you have an areal map then depletion volumes should be represented by meters, unless what they mean is cubic kilometers per square kilometer, but given that the numbers are up to 100 that can't be the case so this figure is undecipherable. In the paper on page 11 the top sentence I see now the authors do indicate it is km3 per grid cell, but their grid cells vary in area across the globe so plotting results in this way is not consistent, nor are the number intuitive. M3/m2 would be much more meaningful. Table 1. The authors report using Sy values of 0.01 and higher for hard rocks, although in the text they way a storage coefficient of 0.001 was used for confined aquifer. Given the storage coefficient is the specific storage (which can vary by a couple of orders of magnitude) times the thickness (also quite variable) it is easy to imagine that the results could be off by a factor of ten. I realize the authors test the sensitivity by varying it by several factors, but given the authors are unable to quantify this uncertainty, it makes their results equally uncertain. Coming up with a "best-fit" scenario does not indicate their parameters are at appropriate values.

Let me summarize be saying there are three main problems I have with this paper: (1) The problem is ill posed. Most abstractions are from confined aquifers, and most water withdrawn from aquifers comes out of storage. Thus estimating the abstractions (or volumes) of withdrawal gives you one estimate (which previous authors have done) and putting those abstractions into a groundwater model does not improve on that—it only gives you head declines. To address depletion globally focus on the regions where the impact is highest. (2) The head declines are only as good as the parameter estimates. Storage coefficients and hydraulic conductivities and thicknesses vary greatly

by magnitude and in space, so the uncertainty in the results will be high, even if you can match some head observations. The fact that in all of the mountainous terrains of the world the simulated depths to water are off by more than an order of magnitude is one warning flag. (3) Their results, in fact, didn't show anything new in terms of depletions. The locations and amounts of depletion are exactly where they told the model the pumping was, and the results were similar to past volume-based results, suggesting the water was mostly coming from storage. In their conclusions the authors state this is only one step in building a better global model, but building a model just for the sake of building it is not scientific progress. This model has an ill-posed application—it is not useful for learning anything new about depletions because the uncertainties are so high. It would be better suited, for example, for looking at how the water table and base flows respond to changes in climate, but it is not clear that the resolution needed for the accuracy for that application is a worthwhile endeavor at the global scale.

---

## Short Comment (SC2) · 3 May 2016

Reviewer 1

We thank Reviewer 1 for the thoughtful and extensive evaluation of our work. He/she raises a number of valid points and concerns, which we hope to address in the answers below. The reviewer's points have certainly been very useful in improving the paper. The reviewer comments are in italics and our rebuttal in roman.

This paper presents an ambitious attempt at hydrologic modeling at the global scale. The work builds on a previous model of the author by adding confined aquifer units and using a transient model. Given the scarcity of physical subsurface data available at the

global scale, large assumptions were made about aquifer structure and parameters. The work is clearly a step in the right direction, and we need to test our ability to model these systems, but the usefulness of the results is not clear.

We thank the reviewer for his/her assessment of the merits of our paper. We agree that large assumptions had to be made. This is yet another step towards a global groundwater model, while we certainly admit that many uncertainties remain.

I have two specific concerns:

1) To calibrate a global model with observations only from the United States and from one delta in Europe doesn't seem reasonable. Especially, given that a major value of the model lies in it's ability to parameterize subsurface systems or predict groundwater level changes for the remainder of the world, where we happen to have few observations. At this point, perhaps the model should just be applied to the US and part of Europe, where the model structure can be better tested?

We appreciate the concerns about the fact that we only used observations from the US and Europe. However, we must remark that an earlier steady-state version of the model (De Graaf et al., 2015) was already validated on all observations provided by Fan et al. (2013), which also consist of observations from Spain, Brazil and Australia. However, many of these observations are not time series but single or time-average numbers. We therefore used these in our previous steady-state version (De Graaf et al., 2015) but not in this transient version. We need time series to calibrate a transient model and these are only freely available in the US and Europe (i.e. in Europe it is not only the delta but the entire Rhine-Meuse basin including Germany, Switzerland and parts of France). We appreciate that we have to extrapolate over the entire earth from this but there are a number of reasons to attempt this: 1) There is no way around it if one seeks to establish a global groundwater model; 2) as stated, the steady state version of the model has been validated against a wider set; 3) Even though only in parts of the world, we use data from large range of climates, geological settings and

landscapes in a split-sample test and are thus reasonably confident that our results make sense; 4) we have supporting validation (from GRACE and reported depletion data from India) that our model produces results are reasonable outside the calibration areas. We will provide an additional discussion on these points in the Discussion part of the paper.

2) I'm concerned about the overall discrepancy between the representative model depth, and the system it aims to simulate. This stems from using surface geology to infer aquifer properties, and similarly, using surface geology to infer the presence and properties of confining units. Many primary aquifer systems are multi-layered with numerous confining units and aquifers with varying properties with depth. The objective of the paper could be to just model the near-subsurface system. However including groundwater extraction values, which in many cases are drawn from deep systems, may force the calibration process beyond reasonable limits.

This is indeed only a second attempt to assemble global hydrogeological model of the world. We know that local groundwater models provide more confining layers in sedimentary basins and we lump these all together in one confining unit and one big aquifer. Using the assumption that productive aquifers coincide with sedimentary basins and sediments below river valleys, the distinction is made between (1) mountain ranges, and (2) sediment basins representing the aquifers. Subsequently, aquifer thicknesses were estimated by relating these to terrain attributes (e.g. curvature) after calibration of these relations with reported aquifer thicknesses from U.S. groundwater modeling studies (de Graaf et al., 2015). These thicknesses are those of productive aquifers until the first impermeable basis. This means that our aquifers lump together aquifers separated by semi-permeable layers into one big aquifer. This may under-estimate head decline. For the confining layer properties those belonging to fine grained or mixed-grained (with and without layers) sediments were taken and for the underlying aquifers the properties belonging to coarse-grained sediments. The storage coefficient of the confined aquifers and the horizontal and vertical conductivities of the confining

layer and aquifers were calibrated. This resulted in an anisotropy ration of kh:kv =10:1 , which effectively corrects for neglecting semi-permeable layers and lumping multi-aquifer systems into one. This is indeed an approximation and will result in a first-order estimate of average head decline over the entire aquifer set. We realize this and hope that by incorporating more regional to local studies in the future our model will slowly grow more mature. We will provide a more thorough discussion of these uncertainties in the Discussion part.

Overall, it's clear that a significant amount of work went into this, and it moves us closer to having a global groundwater model. Addressing some of the comments here about model structure, calibration, and uncertainty in storage change will help clarify the value of the model and it's results.

We thank the reviewer for these nice words and will try to answer the specific questions below. Whenever possible we will add additional explanations and discussion to the manuscript to clarify methods and the value of our results.

Specific comments:

1) It would be helpful to conceptually explain the model and assumptions a bit more clearly in the methods section. Obviously the data required to model these deep aquifers is rare, certainly at the global scale – so the current project is making reasonable assumptions in order to move the understanding forward. Given that, it should be clear early in the paper what system and dynamics it expects to model reasonably well, given the data input restrictions. Broadly, the model improves upon a previous version, which modeled all aquifers as unconfined. Is the current model explicitly modeling the most surficial aquifer and most surficial confining units only? The permeability values represent the surface geology, and the confining unit permeability also seems to be based only on the shallowest layer of material.

See the answer above under item 2. We will more explicitly state in the Introduction what the model supposed to represent and in the discussion what the restriction and

uncertainties are of the approach and what data would be needed to improve estimate in the future.

2) Most of the large-scale (irrigation, industrial, municipal, etc.) groundwater usage is drawn from deep wells, whose regional aquifer characteristics may not be well represented in this model. Can the authors discuss how calibrating the model with relatively shallow aquifer input parameters to fit potentially deep system extraction rates may impact the model performance? It seems like there may be a discrepancy between the system modeled and the one it is calibrated to.

As stated above, we presume to represent the entire stack of upper aquifers until the first impermeable layer (hydrologic basis) without explicitly resolving the semi-permeable layers explicitly but modeling their effects using anisotropic effective conductivity. It means assuming that some surface lithological properties are representative for deeper layers (lacking any other information). But effective properties are calibrated based on the head measurements, some of which are also from deeper sediments. Also, its should be noted that in many areas of the world (India, China, Iran) groundwater abstraction occurs from small agricultural wells that are not deep at all.

3) Given a two-layer model, are interactions with a shallow unconfined aquifer (e.g. alluvial aquifer overlying a confining unit) lost? Are there specific areas where surface water - groundwater dynamics were not well represented, perhaps useful for guiding future research to improve our subsurface parameterization capacity in these areas?

We don't believe so. We estimate the width and depth of the surface water systems using geomorphologic relationships and bankfull discharge. If these dimensions are such that they penetrate the confining layer, which is the case for large rivers and rivers at the edge of coastal confining layers, we have as expected intensive interaction between the surface water system and the aquifer. Otherwise, if the bottom of the surface water is positioned within the confining layers, surface water groundwater interaction is

limited.

4) Were groundwater observations from all well depths used to calibrate the model? For the confined aquifer areas, it is highly possible that groundwater observations are being made in multiple aquifers, where deeper layers would not be expected to have a direct connection with the surface as is being modeled. I understand we cannot expect this level of detail to be included in the model, I'm just curious how fitting a model to these data will impact your results.

We used all the available time series of sufficient length in the calibration. This means indeed that there may be head observations that are not or barely influenced by e.g. surface water levels or recharge while our model does calculate this influence. The lack of information on the precise vertical hydrogeological structure of the aquifers around the observation well screens in this case results in poorer calibration results.

5) The brief description of how aquifer thicknesses were calculated (in addition to the citation to the 2015 paper) is helpful. Can a similar one be provided for how thickness of the confining unit were calculated?

We have added a description of this. For the description of the coastal aquifers we refer to 2.3.2 and for the others (10% of the total thickness) 2.3.3.

6) Were the parameters for the confining units assumed based on the surface unit texture? Were any measurements (or regional model parameters) used to inform individual aquifer confining unit permeability, or were they set uniformly across the globe?

They were set uniformly given the surface texture (fine grained, mixed grained and mix grained layered). Then a single prefactor was used to calibrate them further. So no regional information was used.

7) In the methods section 2.1.2, Does "Next to the river levels" mean proximally adjacent to? Or "next" figuratively? It sound like there are fixed head boundaries being specified at sea level adjacent to all the rivers. If this is correct, can you justify why you

chose to do this? Can you explain this decision with respect to Figure 8? The depth to groundwater appears to follow topography (as you say in the paper), and is simulated quite a bit deeper than observed (e.g. much of western US and Mexico).

This should have stated "apart from the river levels". We will re-write it like that. Furthermore. We state in section 3.3 "Also, for mountain regions deep groundwater tables are simulated. In these areas local aquifers in sedimentary pockets in mountain valleys are smaller than the grid resolution (< 10 km ) and are therefore not captured. As a result, groundwater heads in these regions are likely underestimated (de Graaf et al., 2015)." However, we realize that we should state this more clearly in the introduction already and will do so in the next version of the paper.

8) If it took 10 years for the model to reach equilibrium, does that say something about the degree of disequilibrium in the groundwater system in 1960? Do you think years is reasonable? If so, or not, can you infer something about how the model is functioning?

We don't believe that it is a matter of model functioning. It takes just quite a few years to warm-up the model. This has to do with the large volume of groundwater in the model causing considerable inertia. So we start with a steady state, then run 1960-1970 first and start with 1960 again.

9) There are two periods of rapid groundwater depletion in Figure 12 early 1980s and 2000s. You explain the first as being delayed despite overall abstraction > recharge (is that right?) by stream capture. Is this a process that would be included in the model, without having feedback from groundwater level on surface water?

This process is still there indeed. Although the effect may be somewhat overstated. What happens is that we have the larger rivers connected to the groundwater system in MODFLOW (through the RIV package) and the smaller rivers by the drain package. When one starts to pump more than is being recharged part of it will come out of storage, but in the beginning part will come from reduced discharge (to rivers and drains). After the drains fall dry, part of it may still be supplied by the rivers (river bed infiltration)

and this part increases as the groundwater head drops. After the head drops below the river bottom this infiltration flux becomes constant. So after that most of the additional pumping must come from storage and cause increased depletion rates. Of course, this effect may be overstated because river levels themselves are kept constant in our approach, while they would also decline in reality causing a more gradual increase of depletion rates.

10) The total groundwater depletion is given with 4 significant figures. Can you justify this precision? Can you provide an estimate of uncertainty on the depletion estimate based on errors associated with the groundwater level simulations and storage values?

No we cannot justify this. Thanks for this insight. We should change this to 2 significant figures and use scientific notation.

11) The conclusion that model performance is only slightly better with the inclusion of the confined systems suggest that we do not need to model the confined systems? Or that they should be modeled another way?

That is a good point. The model performance in terms of heads is not better. However, our estimates in terms of depletion rates are closer to previous volume-based estimates (e.g. Konikow, 2011) and thus believe that these are better. We will add this observation to the paper.

12) Many of the figures can be tightened up: they could use subfigure labels (A,B,C,etc.), and make sure the axis labels are final (some say "data", several are missing), and that for figures with subplots that the axis line up for all figs. There are a handful of typos, but those can be corrected with minor effort.

We will provide updated figures in the next version.

---

## Short Comment (SC3) · 3 May 2016

Reviewer 2

We thank Reviewer 2 for the extensive evaluation of our work. He/she provides several reasons why the model should not have been made and why it does not add to previous work on global depletion. We disagree with this assessment obviously and we agree with some of his/her critique, but disagree with most of it as shown hereafter.

We will start with the main points summarized at the end of the review and then address any other specific questions and remarks that have been made above.

1) The problem is ill posed. Most abstractions are from confined aquifers, and most

water withdrawn from aquifers comes out of storage. Thus estimating the abstractions (or volumes) of withdrawal gives you one estimate (which previous authors have done) and putting those abstractions into a groundwater model does not improve only gives you head declines. To address depletion globally focus on the regions where the impact is highest.

We disagree with this assessment. Yes, many abstractions are from confined aquifers but it does not mean that there isn't any connection with the surface water system when abstracting water from a confined aquifer. A well pumping from an unconfined aquifer will tend to capture a considerable part of its discharge from the nearest steam. The presence of a confining layer between the well and the stream causes the cone of depression to extend to greater distances to capture the natural discharge required to offset pumping (e.g. Morgan and Jones 1999). Some of the current authors also authored previous assessment of global depletion (Wada et al., 2010) and the big critique of this work was that we did not account for increased capture an thus over-estimated depletion. By using a groundwater flow model we now do take account of this. This can be seen from Figure 12 that clearly shows a big difference between depletion rates (water abstracting from storage) calculated with the groundwater model and those simply obtained from abstraction minus recharge. See also the difference between Wada's estimate and ours (Table 3) where ours is lower due to taking account of increased capture.

(2) The head declines are only as good as the parameter estimates. Storage coefficients and hydraulic conductivities and thicknesses vary greatly by magnitude and in space, so the uncertainty in the results will be high, even if you can match some head observations. The fact that in all of the mountainous terrains of the world the simulated depths to water are off by more than an order of magnitude is one warning flag.

We agree totally that the head declines are as good as the parameter estimates. In general this is true for all models. The question then is: should we or should we not attempt to model heads and head declines if we know that uncertainty is high? Our

answer is clearly "yes", provided that you show this uncertainty (and we do: see Figures 6, 7 and 13). If you do not model it at all, you will never discover how uncertain one really is and which data are lacking and where. Moreover, if you model a system, even if its parameters are subject to uncertainty, one can still learn a lot from exploring the system behaviour and system sensitivities. Finally, regarding the remarks about the (too) deep groundwater levels in mountains. We explicitly state that our model does not resolve the perched water tables in hillslopes and the water tables in alluvial pockets in mountain valleys (these are conceptually modeled in the surface water model), but calculate the groundwater in the bedrock below (we will state this earlier in the Introduction in a second version). This is the cause of the underestimation in the mountains. However, this does by no means disqualify our results in the larger alluvial basins coastal plains and deltas where groundwater is relatively shallow, groundwater depth matters and is often being depleted.

(3) Their results, in fact, didn't show anything new in terms of depletions. The locations and amounts of depletion are exactly where they told the model the pumping was, and the results were similar to past volume-based results, suggesting the water was mostly coming from storage. In their conclusions the authors state this is only one step in building a better global model, but building a model just for the sake of building it is not scientific progress. This model has an ill-posed application. It is not useful for learning anything new about depletions because the uncertainties are so high. It would be better suited, for example, for looking at how the water table and base flows respond to changes in climate, but it is not clear that the resolution needed for the accuracy for that application is a worthwhile endeavor at the global scale.

We agree partly about what this paper adds to previous studies. No, it does not reveal new areas with groundwater depletion. This makes sense as the groundwater abstractions used here are closely related to those used by Wada et al. (2010), apart from that they are not imposed from the IGRAC data, but simulated by the surface water model itself (according to De Graaf et al., 2013). However, the depletion rates

are different and reveal, albeit with considerable uncertainty, the effect of increased capture (see our response to reviewer 1 for the mechanism). Moreover, and this is also the answer to the first point that was raised at the beginning of the review (see hereafter), calculating depletion volumes is an important application of this model, but not the only one. There are multiple reasons for building a groundwater model instead of using a single storage-outflow reservoir in land surface models: 1) Indeed, also in case of confined aquifers there is considerable interaction between surface water and groundwater. This is difficult to parameterize in a land-surface model. 2) Our ultimate goal is also to estimate head declines. This is important because one needs to know how the depth-to-groundwater develops with time. If the groundwater head is lower than 100 below the surface ordinary farmers' pumps cannot reach it anymore and if it is lower than 300 m it becomes unattainable for ordinary industrial pumps as well. We accept that these estimates will be subject to considerable uncertainty. But so are the estimates of depletion rates. 3) Even though the volumes of groundwater traveling across catchment boundaries is limited, it becomes more and more important if one goes to even higher resolutions, and certainly in case of groundwater abstractions. Thus, our work already prepares for this. 4) In many areas of the world groundwater significantly contributes to evaporation through groundwater convergence and capillary rise (see Fan et al., 2007; Bierkens and Van den Hurk, 2009) which warrants modeling groundwater head explicitly. So in conclusion: this study adds considerably to previous depletion estimates and provides the first global estimates of global groundwater head variability and head decline and is a first step toward assessing how long our groundwater reserves will technically and economically last.

Hereafter we answer other specific points raised by the reviewer:

Page 1. Line 1. This sentence appears to be the justification for building a global groundwater model, but this sentence is misleading. If by basins they mean surface water basins, then lateral flow is only significant in the most surficial part of the system where >90% of the flow occurs. Most substantial groundwater withdrawals are from

deeper confined and semi-confined aquifers where the total natural flow in the system is very small compared to the recharge, for example. It is true that most hydrological models do not include a groundwater flow component, but that in itself does not justify building a global groundwater model. Basically as the saying goes, "all models are wrong, but some are more useful than others"–so the authors have to demonstrate that their model is both accurate "enough" and "useful" to some degree. Unfortunately don't believe this model is either. It is only in the ballpark where the data exist in the 10% of the world where most of the pumping is occurring (it does a very bad job in much of the world in simply fitting depth to the water table), and it really isn't useful in the 10% of the world where the pumping is occurring because global groundwater depletion has already been quantified to a similar degree of accuracy (Wada et al, 2010, 2012; Konikow, 2011), and the groundwater model they are constructing really doesn't improve on those estimates. Lately it seems to be that the bigger the model is, the better it must be and the more likely we are to learn something new from it. Therefore the authors are now producing a global model; but in this case the model does not produce any useful results. More on this later.

We agree with the reviewer that, albeit flowpath analysis (See Figure 9 and also De Graaf et al., 2015) shows that many flowpaths passing catchment boundaries, the associated volumes are limited. However, we also observe a trend that future land surface models will operate at increasingly higher resolution (Bierkens et al., 2014), which makes that the across boundary fluxes at the grid scale will become more and more important, particular in case of groundwater abstractions. Our modeling effort is part of preparing for that circumstance. Also, as argued above (point 3) there are many other reasons for wanting to simulate groundwater heads at larger scales. We will more explicitly provide these reasons in our revised version

Page 1, line 19. This has been known for decades-why cite a reference from 2016? Why not a UN report from the 20th century?

The reviewer is right and we will add a reference to that effect.

Page 2, line 29. Here the problem begins. Indeed most abstractions occur in confined aquifers (which is why the investigators include a second layer to represent them), but simulating head declines there does NOT improve any estimates of groundwater depletion. Most all groundwater depletion in deep aquifers comes out of storage, not from bounding surface-water bodies, and therefore the model -simulated depletions only mimic the abstractions that are put into the model a priori. Thus the model is only as good as the estimate of water abstractions that are available. The earlier global depletion studies did their best to estimate these abstractions. Putting these numbers into a groundwater model does not improve on them.

As argued above under point 1), it is not true that most of the water comes out of storage. Many of the abstractions in our model indeed are from confined aquifers, but they certainly attract groundwater at the expense of baseflow or river runoff as shown in Figure 12 and Table 3.

Figure 5 horizontal axes need labels (year). "Data" is not an acceptable y-axis labels aren't these heads, in meters?

The reviewer is right. We apologize for the oversight and will change this in the revised version.

Figure 6 is very hard to understand because the x-axes are not labeled and the y-axes are underlabeled. Count of what?

We will put the required labels in these figures in the revised version.

Figure 7. The points are too faint. It's not clear how far the simulated heads actually fall below the measured heads. Axis labels need to include "in meters".

The points have been deliberately plotted a bit faint to show the effects of point density in the Figure. These are thousands of points, so it is difficult to see individual points back. We will try to improve and put (in m) on the axes.

Figure 8. These results truly show how bad this model is. Only in the deserts of the

world is the depth to the water table typically greater than 100 m. Yet most of the mountain ranges here show depths greater than 300 m. There are streams in most of those mountains so we know the depth to water is zero at those locations and likely a few tens of meters at the watershed divides. If so much of the world gives results this bad it does not give me confidence in the rest of the world, and there really isn't any reason to do a model of the whole world, especially when 90% of the abstraction occurs in 10% of the area. The authors will argue that these areas don't have any significant depletion anyway, but that argues that those areas really don't need to be modeled. If the time and resources were focused on the 10% of the world where depletion was really occurring, you would get better results and might learn something.

As argued under point 2 above, we clearly state that we are not able to simulate the perched water tables in hillslopes and the water tables in alluvial pockets in mountain valleys (We hope to resolve these in the near future), but calculate the groundwater in the bedrock below. The conclusion that as a result the results should be therefore bad in areas where groundwater tables are shallow is thus not at all founded, and we have the validation results to show that we do a good job there. We do not think it is a good idea to limit the model to areas where there is depletion because: 1) we expect other areas that have yet no depletion to develop this in the future (parts of Africa, especially cities in deltas); 2) as stated under point 3 above, the groundwater model has other purposes than calculating depletion rates. Finally, in a previous paper (De Graaf et al., 2015) about the one-layer steady state version of the model we have masked the mountain areas out. We elected not to do this here, but could do this again.

Figure 11. This figure makes no sense. If you have an areal map then depletion volumes should be represented by meters, unless what they mean is cubic kilometers per square kilometer, but given that the numbers are up to 100 that can't be the case so this figure is undecipherable. In the paper on page 11 the top sentence I see now the authors do indicate it is km3 per grid cell, but their grid cells vary in area across the globe so plotting results in this way is not consistent, nor are the number intuitive.

M3/m2 would be much more meaningful.

This is a valid point: we will change this to M3/m2 in a revised version.

Table 1. The authors report using Sy values of 0.01 and higher for hard rocks, although in the text they way a storage coefficient of 0.001 was used for confined aquifer. Given the storage coefficient is the specific storage (which can vary by a couple of orders of magnitude) times the thickness (also quite variable) it is easy to imagine that the results could be off by a factor of ten. I realize the authors test the sensitivity by varying it by several factors, but given the authors are unable to quantify this uncertainty, it makes their results equally uncertain. Coming up with a "best-fit" scenario does not indicate their parameters are at appropriate values.

We are uncertain what we need to do with this remark. We calibrated the model to fit the head time series and we show the uncertainty in Figure 4. We will discuss the uncertainty more extensively in the discussion. We will refer to De Graaf et al (2015) where we already investigated the coefficient of variation of results (a measure of uncertainty) as a result of both conductivity and aquifer thickness uncertainty.

---

## Author Comment (AC1) · 6 Jun 2016

Reply to comments by Petra Döll

We thank Petra Döll for the thoughtful and extensive evaluation of our work. She raises a number of valid points, additions and concerns, which we hope to address in the answers below. The reviewer's points will certainly be very useful in improving the paper.

Gradient-based groundwater modeling at the global scale is a big challenge, and it was very interesting to me to see results of such an effort in the manuscript of de Graaf et al. I would like to make some comments and pose questions regarding 1) the

groundwater modeling and its presentation in general (as there are a few aspect that need clarification) and 2) the estimation of groundwater depletion.

(1) Global groundwater modeling in general

Page 5, line 20: Regarding discharge of groundwater to local springs etc. From the wording, it is not clear if water storage S3 is from PCR-GLOBWB or the elevation of the floodplain. Please clarify. How is the elevation of the floodplain determined, and where is it as compared to DEM, HRIV and RBOT? Please indicate in the manuscript how large this local drainage component is compared to total Qbf (globally).

The water storage S3 is indeed taken from the original version of PCR-GLOBWB (the third reservoir). From the DEM and the storage in S3 (aquifer depth times porosity) we can calculate the amount of storage [L] above the floodplain elevation. From this follows the local drainage flux. Globally this is 65% of the total baseflow. We will add these descriptions to the manuscript.

Page 8, Line 4: If groundwater head falls below the bottom of the confining layer, is then the confined aquifer modelled as unconfined, and the storage coefficient set to the specific yield to Table 1 instead of 0.001?

Ideally it would, but our approach does not allow this at this time. Running MODFLOW with the option of cells being wetted and rewetted is not feasible due to convergence issues. This would result in too many outer iterations in the solver and consequently very long run times (already three weeks currently). Changing the parameters from confined to unconfined will affect the depletion estimate.

Fig. 5: Typo in figure caption: Central Valley is unconfined, the Netherland confined.

Thank you we will correct this.

Fig. 6: I was wondering whether all three graphs showing R and all three graphs showing QRE really have the same y-axis (they should!). The units for QRE are %? Regarding QRE, model performance decreases according to Fig. 6, while the two

examples in Fig. 5 show clearly the advantage of distinguishing between confined and unconfined aquifers, for the two wells shown. Why is this so? Can you localize where, in the confined parts of your model, assuming unconfined conditions leads to a better modeling of head amplitudes? I think it is necessary to include, in the conclusions (p. 14, l 12-14), that including confining layers leads to a worse simulation of head variations.

This is a valid point. We will add a sentence stating where the inclusion of the confining layer increases performance and where it decreases. We will highlight the areas where groundwater dynamics are not well represented and point out which research is needed to improve the model outputs in future. Also, we will replace "count" with "relative frequency" in the next version of the manuscript.

Fig. 9: In my understanding, Fig. 9 does not show flow paths but travel times. What exactly is shown? The travel time of the groundwater recharged at the grid cell shown? Please also clarify in the text on p. 12. I also think that your conclusion (page 14, lines 18-20) is not backed by Figs. 9 or 10. Fig. 9 shows shorter travel times in case of confined aquifer modeling, while Fig. 10 shows significant importing/exporting in mountainous areas (not flat confined areas). (Units are missing on y-axis of Fig. 10).

The figure shows both. It shows the length of the flow path and the apparent age of the groundwater along the flow paths. The flow path is calculated by releasing a particle in a cell at the location of the groundwater table and then following it (using MODPATH) to a (weak) sink. We will put a clear description in our revised manuscript. Regarding the conclusions: yes, they are not correct. We see long flow paths in case of confining layers, but not slow. Indeed travel times are shorter, so we will remove the word "slow". Regarding exporting and importing we can see in Fig. 10 that in the confined scenarios (e.g. a confined layer for the High Plain aquifer) larger- values importers and exporters are simulated compared to the unconfined scenario. This is reflected by the longer flowpaths simulated in the confined scenario crossing catchment boundaries

(2) Simulation of groundwater depletion

Page 13, lines 20-25: Please define "volume-based" and "flux-based" approaches. Do you compute groundwater depletion by subtracting heads and multiplying with the storage coefficient? Do you call your approach "volume-based? Otherwise, I would not agree with the sentence: "Figure 14 (Online Supplementary Information) shows that effect of hydraulic properties on the groundwater depletion volumes (note volumes, not heads) is considerable, which makes estimates of groundwater depletion by volume based methods rather uncertain."

Indeed, our approach is volume-based as we interpret this to mean calculating the actual volume taken out of storage (this is in our approach change in head times storage coefficient or specific yield). Flux-based approaches are the ones obtained by calculating depletion as the difference between recharge and abstraction. Volume-based approaches are certainly dependent on hydraulic properties: storage coefficient or specific yield relating head change to volume taken out of storage, but also the head change itself which depends on hydraulic conductivity and the river bed conductance of any river in the vicinity of dropping groundwater heads. In Figure 14 we show the model sensitivity using different parameter sets (listed in Table 2).

The development of the global sum of groundwater storage loss in cells in which net recharge is smaller than groundwater abstractions (as shown in Figs. 12 and 14) shows how sensitive this estimate is to parameter settings. If depletion assuming unconfined conditions only is (in 2005-2010 compared to 1960) twice the amount of that for confined conditions, I would think the model is overly sensitive. Equally important, Fig. 12, with an increase in groundwater storage before 1980 as compared to 1960, and an actual onset of groundwater depletion only in 1998 indicates to me that what you see in the first decades may be caused by the fact that 10 years of running 1960 (climate and water abstractions) on steady-state groundwater levels was not enough to get a reasonable situation of the state of groundwater heads in depletion areas in 1960. Or that the location of the groundwater table to which interaction with surface water is very

sensitive was not close enough to reality (see Fig. 7 where e.g. differences of 20 m to observations are the rule but that would already have a strong impact on gw-sw interactions). How did the flows between rivers and groundwater develop over time in the depletion areas? Regarding the temporal dynamics, your simulation results show a large depletion of 5000 km3 in only 6 years (1998-2004), with relatively little dynamic at other times. Is this due to spatial averaging?

You may certainly have a point here. The delayed response of global depletion is understandable as a result of the groundwater-surface water interaction. As we answered to questions of reviewer 1 related to the late response: "This process is still there indeed. Although the effect may be somewhat overstated. What happens is that we have the larger rivers connected to the groundwater system in MODFLOW through the RIV package, and the smaller rivers by the drain package. When one starts to pump more than is being recharged part of the abstraction will come out of storage, but in the beginning part will come from reduced discharge (to rivers and drains) too. After the drains fall dry, part of the abstraction may still be supplied by the rivers (river bed infiltration) and this part increases as the groundwater head drops. After the head drops below the river bottom this infiltration flux becomes constant. So after that most of the additional pumping must come from storage and cause increased depletion rates. Of course, this effect may be overestimated because river levels are simulated by PCR-GLOBWB and are not fully coupled to MODFLOW yet. River levels could decline more in reality causing a more gradual increase of depletion rates." However, we agree that this process and the time of disattachment of groundwater and surface water may be quite sensitive to the initial conditions. So a longer warm-up period may be a good idea, for the revised manuscript we will do so (e.g. 50 years).

In page 4, line 27 you state that after 10 years of running 1960, a dynamic equilibrium was reached? Do you mean that after 10 years, groundwater heads did not change by +- x%? What value did you choose for x? As a test for sensitivity of Fig. 12, I suggest you rerun your model with 100 years of 1960 initialization instead of 10 years,

and show the results in Fig. 14.

We agree that a longer warm-up period is a good idea. We will do so the revised paper. (See also previous comment.)

Another question regarding Fig. 14: The grey recharge-abstraction curve is more or less a straight line, e.g. the annual difference between groundwater recharge and abstraction is constant between 1960 and 2010. Can you explain why, as groundwater withdrawals are known to have increased significantly during these fifty years?

We see that something is wrong here, and the reviewer is totally right as this point. We found out that in this figure the plotted line of the abstraction-recharge deficit (abst-rch) presents wrong information; a mistake was made in summing up total recharge rates. This concerns the postprocessing and not in the actual calculation, and thus it does not change the depletion estimates of the maximum and unconfined scenario (also presented in Figure 12). We apologize for this mistake and will include the updated the figure shown in the attachment.

This figure shows an exponential curve for the abstraction-recharge deficit, which is inline what we expect as groundwater abstractions are indeed increasing over 1960-2010. Our estimate is within the same range as previous flux based estimates, e.g. this studies 2000 deficit is ~19400 km3 is comparable to the estimate of Pokharel (2000);~19000 km3. We will rewrite the discussion of this figure in the manuscript (p.13, l.12-l20).

In Fig. 13 C, please add observations for High Plains Aquifer, while in Fig. 13A I suggest you use the same legend/color as in 13B, for better comparability.

We will do so.

In Fig. 11, use mm instead of km3.

We will do so.

In your comparison to other estimates of global groundwater depletion, please add a comparison to the maps and global values of Döll, P., Muller Schmied H., Schuh, C., Portmann F. T., Eicker A. (2014): Global-scale assessment of groundwater depletion and related groundwater abstractions: Combining hydrological modeling with information from well observations and GRACE satellites, Water Resour. Res., 50 (7), 5698-5720, doi:10.1002/2014WR015595. In this study, we took into account that baseflow is reduced (and then zero) in areas of groundwater depletion (but inflow of river water into groundwater is not simulated). Also, there is groundwater recharge from surface water bodies in dry areas (very rough estimate).

We will add this to our comparison. The effects conceptually modeled in your approach are implicitly accounted for when using a global groundwater flow model, although the effect of increased capture may be over-estimated in our case as we do not have a full coupling allowing for a decline in surface water levels as a result of groundwater pumping. We are preparing a paper with fully coupled groundwater-surface water interaction that does allow falling river levels as a result of groundwater pumping.

In addition we assumed, based on comparing our modeling results to many independent estimates including GRACE, that 70% deficit irrigation is done in groundwater depletion areas. This resulted in a best estimate of global groundwater depletion of 2240 km3 for 1960-2000, and of 3257 km3 for 1960-2009. So according to your study, gw depletion increases much faster after 2000 than in our study, and the total value is higher.

Indeed, we assumed optimal irrigation volumes that may explain the larger depletion rates found in our study.

[Figure]

Figure 12: Global depletion trends simulated for the maximum (max. confined) and unconfined
aquifer scenarios and compared to the estimated cumulative deficit between simulated recharge
and groundwater abstraction (abst-rch).

**Fig. 1.**

---

## Author Comment (AC2) · 6 Jun 2016

Reviewer 1

We thank Reviewer 1 for the thoughtful and extensive evaluation of our work. He/she raises a number of valid points and concerns, which we hope to address in the answers below. The reviewer's points will certainly contribute to the improving our manuscript.

This paper presents an ambitious attempt at hydrologic modeling at the global scale. The work builds on a previous model of the author by adding confined aquifer units and using a transient model. Given the scarcity of physical subsurface data available at the global scale, large assumptions were made about aquifer structure and parameters.

The work is clearly a step in the right direction, and we need to test our ability to model these systems, but the usefulness of the results is not clear.

We thank the reviewer for his/her assessment of the merits of our paper. We agree that large assumptions had to be made. This is yet another step towards a global groundwater model, while we admit that many uncertainties remain.

I have two specific concerns:

1) To calibrate a global model with observations only from the United States and from one delta in Europe doesn't seem reasonable. Especially, given that a major value of the model lies in it's ability to parameterize subsurface systems or predict groundwater level changes for the remainder of the world, where we happen to have few observations. At this point, perhaps the model should just be applied to the US and part of Europe, where the model structure can be better tested?

We appreciate the concerns about the fact that we only used observations from the US and Europe. However, we must remark that an earlier steady-state version of the model (De Graaf et al., 2015) was already validated on all observations provided by Fan et al. (2013), which also consist of observations from Spain, Brazil and Australia. However, many of these observations are not time series but single or time-average values. We therefore used these in our previous steady-state version (De Graaf et al., 2015) but not to validate the present transient simulation. The required time series to calibrate a transient model are only freely available in the US and Europe (note that in Europe not only data from the delta is used, but the entire Rhine-Meuse basin including Germany, Switzerland and parts of France). We appreciate that we have to extrapolate our model over the most of the earth but this current attempt can be motivated by the following reasons: 1) as stated, the steady state version of the model has been validated against a wider set; 2) Even though only in parts of the world, we use data from large range of climates, geological settings and landscapes in a split-sample test and are thus reasonably confident about the validation of the model; 3)

[Figure]

we have supporting validation (from GRACE and reported depletion data from India) that our model produces results are reasonable outside the calibration areas; 4) There is no way around this data lack if one seeks to establish a global groundwater model. We will provide an additional discussion on these points in the Discussion part of the paper.

2) I'm concerned about the overall discrepancy between the representative model depth, and the system it aims to simulate. This stems from using surface geology to infer aquifer properties, and similarly, using surface geology to infer the presence and properties of confining units. Many primary aquifer systems are multi-layered with numerous confining units and aquifers with varying properties with depth. The objective of the paper could be to just model the near-subsurface system. However including groundwater extraction values, which in many cases are drawn from deep systems, may force the calibration process beyond reasonable limits.

This is indeed only a second attempt to assemble global hydrogeological model of the world. We know that local groundwater models provide more confining layers in sedimentary basins and we lump these all together in one confining unit and one big aquifer. Using the assumption that productive aquifers coincide with sedimentary basins and sediments below river valleys, the distinction is made between (1) mountain ranges, and (2) sediment basins representing the aquifers. Subsequently, aquifer thicknesses were estimated by relating these to terrain attributes (e.g. curvature) after calibration of these relations with reported aquifer thicknesses from U.S. groundwater modeling studies (de Graaf et al., 2015). These thicknesses are those of productive aquifers until the first impermeable basis. This means that our aquifers lump together aquifers separated by semi-permeable layers into one big aquifer system. This may under-estimate head decline. To the confining layer, properties belonging to fine grained or mixed-grained sediments (with and without layers) were assigned and the properties belonging to coarse-grained sediments to the underlying aquifer. The storage coefficient of the confined aquifers and the horizontal and vertical conductivities of the confining layer and

aquifers were calibrated. This resulted in an anisotropy ration of kh:kv =10:1 , which effectively corrects for neglecting semi-permeable layers and lumping multi-aquifer systems into one. This is indeed an approximation and will result in a first-order estimate of average head decline over the entire aquifer set. We realize this and hope that by incorporating more regional to local studies in the future our model will slowly grow more mature. We will provide a more thorough discussion of these uncertainties in the Discussion part.

Overall, it's clear that a significant amount of work went into this, and it moves us closer to having a global groundwater model. Addressing some of the comments here about model structure, calibration, and uncertainty in storage change will help clarify the value of the model and it's results.

We thank the reviewer for the encouragement and will try to answer the specific questions below. Whenever possible we will add additional explanations and discussion to the manuscript to clarify methods and the value of our results.

Specific comments:

1) It would be helpful to conceptually explain the model and assumptions a bit more clearly in the methods section. Obviously the data required to model these deep aquifers is rare, certainly at the global scale – so the current project is making reasonable assumptions in order to move the understanding forward. Given that, it should be clear early in the paper what system and dynamics it expects to model reasonably well, given the data input restrictions. Broadly, the model improves upon a previous version, which modeled all aquifers as unconfined. Is the current model explicitly modeling the most surficial aquifer and most surficial confining units only? The permeability values represent the surface geology, and the confining unit permeability also seems to be based only on the shallowest layer of material.

See the answer above under item 2. We will more explicitly state in the Introduction what the model supposed to represent and in the discussion what the restriction and

uncertainties are of the approach and what data would be needed to improve estimate in the future.

2) Most of the large-scale (irrigation, industrial, municipal, etc.) groundwater usage is drawn from deep wells, whose regional aquifer characteristics may not be well represented in this model. Can the authors discuss how calibrating the model with relatively shallow aquifer input parameters to fit potentially deep system extraction rates may impact the model performance? It seems like there may be a discrepancy between the system modeled and the one it is calibrated to.

As stated above, we presume to represent the entire stack of upper aquifers until the first impermeable layer (hydrologic basis) without explicitly resolving the semi-permeable layers explicitly but modeling their effects using anisotropic effective conductivity. It means assuming that some surface lithological properties are representative for deeper layers (lacking any other information). But effective properties are calibrated based on the head measurements, some of which are also from deeper sediments. Also, its should be noted that in most areas of the world groundwater abstraction occurs from small agricultural wells that are not deep at all.

3) Given a two-layer model, are interactions with a shallow unconfined aquifer (e.g. alluvial aquifer overlying a confining unit) lost? Are there specific areas where surface water - groundwater dynamics were not well represented, perhaps useful for guiding future research to improve our subsurface parameterization capacity in these areas?

We estimate the width and depth of the surface water systems using geomorphologic relationships and bankfull discharge. If these dimensions are such that they penetrate the confining layer, which is the case for large rivers and rivers at the edge of coastal confining layers, we have, as expected, intensive interaction between the surface water system and the aquifer. Otherwise, if the bottom of the surface water is positioned within the confining layers, surface water groundwater interaction is limited.

In de revised manuscript we will add a discussion where the inclusion of confining

layers increases model performance and where it decreases (see also the points razed by Petral Döll about Fig 6). We will highlight the areas where groundwater dynamics are not well represented and point out which research is needed to improve the model outputs in future.

4) Were groundwater observations from all well depths used to calibrate the model? For the confined aquifer areas, it is highly possible that groundwater observations are being made in multiple aquifers, where deeper layers would not be expected to have a direct connection with the surface as is being modeled. I understand we cannot expect this level of detail to be included in the model, I'm just curious how fitting a model to these data will impact your results.

We used all the available time series of sufficient length in the calibration. This means indeed that there may be head observations that are not or barely influenced by e.g. surface water levels or recharge while our model does calculate this influence. The lack of information on the precise vertical hydrogeological structure of the aquifers around the observation well screens in this case results in poorer calibration results.

5) The brief description of how aquifer thicknesses were calculated (in addition to the citation to the 2015 paper) is helpful. Can a similar one be provided for how thickness of the confining unit were calculated?

We will add a description of this in the Methods. For the description of the coastal aquifers we refer to 2.3.2 and for the others (10% of the total thickness) 2.3.3.

6) Were the parameters for the confining units assumed based on the surface unit texture? Were any measurements (or regional model parameters) used to inform individual aquifer confining unit permeability, or were they set uniformly across the globe?

They were set uniformly given the surface texture (fine grained, mixed grained and mix grained layered). Then a single prefactor was used to calibrate them further. So no regional information was used.

7) In the methods section 2.1.2, Does "Next to the river levels" mean proximally adjacent to? Or "next" figuratively? It sound like there are fixed head boundaries being specified at sea level adjacent to all the rivers. If this is correct, can you justify why you chose to do this? Can you explain this decision with respect to Figure 8? The depth to groundwater appears to follow topography (as you say in the paper), and is simulated quite a bit deeper than observed (e.g. much of western US and Mexico).

This should have stated "apart from the river levels". We will re-write it like that. Furthermore. We state in section 3.3 "Also, for mountain regions deep groundwater tables are simulated. In these areas local aquifers in sedimentary pockets in mountain valleys are smaller than the grid resolution (< 10 km ) and are therefore not captured. As a result, groundwater heads in these regions are likely underestimated (de Graaf et al., 2015)." However, we realize that we should state this more clearly in the introduction already and will do so in the next version of the paper.

8) If it took 10 years for the model to reach equilibrium, does that say something about the degree of disequilibrium in the groundwater system in 1960? Do you think years is reasonable? If so, or not, can you infer something about how the model is functioning?

We don't believe that it is a matter of model functioning. It takes just quite a few years to warm-up the model. This has to do with the large volume of groundwater in the model causing considerable inertia. So we start with a steady state, then run 1960-1970 first and start with 1960 again. In the revised manuscript we will increase the spin-up period and analyze the effect of this increased spin-up period.

9) There are two periods of rapid groundwater depletion in Figure 12 early 1980s and 2000s. You explain the first as being delayed despite overall abstraction > recharge (is that right?) by stream capture. Is this a process that would be included in the model, without having feedback from groundwater level on surface water?

This process is still there indeed. Although the effect may be somewhat overstated. What happens is that we have the larger rivers connected to the groundwater system

in MODFLOW (through the RIV package) and the smaller rivers by the drain package. When one starts to pump more than is being recharged part of it will come out of storage, but in the beginning part will come from reduced discharge (to rivers and drains). After the drains fall dry, part of it may still be supplied by the rivers (river bed infiltration) and this part increased as the groundwater head drops. After the head drops below the river bottom this infiltration flux becomes constant. So after that most of the additional pumping must come from storage and cause increased depletion rates. Of course, this effect may be overestimated because river levels are simulated by PCR-GLOBWB and are not fully coupled to MODFLOW yet. River levels could decline more in reality causing a more gradual increase of depletion rates.

10) The total groundwater depletion is given with 4 significant figures. Can you justify this precision? Can you provide an estimate of uncertainty on the depletion estimate based on errors associated with the groundwater level simulations and storage values?

No we cannot justify this. Thanks for this insight. We will change this to 2 significant figures and use scientific notation.

11) The conclusion that model performance is only slightly better with the inclusion of the confined systems suggest that we do not need to model the confined systems? Or that they should be modeled another way?

That is a good point. The model performance in terms of heads is not better. However, our estimates in terms of depletion rates are closer to previous volume-based estimates (e.g. Konikow, 2011) and thus believe that our new simulation captures the surface-groundwater interactions more realistically. We will add this observation to the paper.

12) Many of the figures can be tightened up: they could use subfigure labels (A,B,C,etc.), and make sure the axis labels are final (some say "data", several are missing), and that for figures with subplots that the axis line up for all figs. There are a handful of typos, but those can be corrected with minor effort.

We will provide updated figures in the next version.

---

## Author Comment (AC3) · 6 Jun 2016

Reviewer 2

We thank Reviewer 2 for the extensive evaluation of our work. He/she provides several reasons why the model should not have been made and why it does not add to previous work on global depletion. Obviously we disagree with this assessment, but do take some of his/her critique at heart.

We will start with the main points summarized at the end of the review and then address any other specific questions and remarks that have been made above.

(1) The problem is ill posed. Most abstractions are from confined aquifers, and most

water withdrawn from aquifers comes out of storage. Thus estimating the abstractions (or volumes) of withdrawal gives you one estimate (which previous authors have done) and putting those abstractions into a groundwater model does not improve only gives you head declines. To address depletion globally focus on the regions where the impact is highest.

We disagree with this assessment. Yes, many abstractions are from confined aquifers but it does not mean that there isn't any connection with the surface water system when abstracting water from a confined aquifer. A well pumping from an unconfined aquifer will tend to capture most of its discharge from the nearest steam. The presence of a confining layer between the well and the stream causes the cone of depression to extend to greater distances to capture the natural discharge required to offset pumping (e.g. Morgan and Jones 1999). Some of the current authors also authored previous assessment of global depletion (Wada et al., 2010) and the main critique of this work was not accounting for increased capture and thus over-estimated depletion. By using a groundwater flow model we now do take account of this increased capture. This is illustrated by Figure 12 that clearly shows a big difference between depletion rates (water abstracting from storage) calculated with the groundwater model and those simply obtained from abstraction minus recharge. See also the difference between Wada's estimate and ours (Table 3) where ours is lower due to taking account of increased capture.

(2) The head declines are only as good as the parameter estimates. Storage coefficients and hydraulic conductivities and thicknesses vary greatly by magnitude and in space, so the uncertainty in the results will be high, even if you can match some head observations. The fact that in all of the mountainous terrains of the world the simulated depths to water are off by more than an order of magnitude is one warning flag.

We agree that the head declines are as good as the parameter estimates. In general this is true for all models. In line with good modeling practice we present our results with clear bounds of uncertainty (see Figures 6,7, and 13). The simulation and associated

uncertainty are indicative of the sensitivity of the model and of the directions in which improvements must be sought. These improvements are discussed in the discussion. If you do not model e.g. groundwater flows or depletion at all, you will never discover how uncertain one really is and which data are lacking and where. Moreover, if you model a system, even if its parameters are subject to uncertainty, one can still learn a lot from exploring the system behaviour and system sensitivities. Finally, regarding the remarks about the (too) deep groundwater levels in mountains. We explicitly state that our model does not resolve the perched water tables in hillslopes and the water tables in alluvial pockets in mountain valleys (these are conceptually modeled in the surface water model), but calculate the groundwater in the bedrock below. This is the cause of the underestimation in the mountains. We will state this earlier in the Introduction in the revised version of the manuscript to be more clear about this point. However, this does by no means disqualify our results in the larger alluvial basins coastal plains and deltas where groundwater is relatively shallow, groundwater depth matters and where groundwater is often being depleted due to the high demand.

(3) Their results, in fact, didn't show anything new in terms of depletions. The locations and amounts of depletion are exactly where they told the model the pumping was, and the results were similar to past volume-based results, suggesting the water was mostly coming from storage. In their conclusions the authors state this is only one step in building a better global model, but building a model just for the sake of building it is not scientific progress. This model has an ill-posed application. It is not useful for learning anything new about depletions because the uncertainties are so high. It would be better suited, for example, for looking at how the water table and base flows respond to changes in climate, but it is not clear that the resolution needed for the accuracy for that application is a worthwhile endeavor at the global scale.

We concede that our study does not reveal new areas of groundwater depletion. This makes sense as the groundwater abstractions used here are closely related to those used by Wada et al. (2010), apart from that they are not taken directly from the IGRAC

data but total demand is allocated to the groundwater and surface water resource on the basis of availability (according to De Graaf et al., 2013). However, the depletion rates are different and reveal, albeit with considerable uncertainty, the effect of increased capture (see our response to reviewer 1 for the mechanism). Moreover, and this is also the answer to the first point that was raised at the beginning of the review (see hereafter), calculating depletion volumes is an important application of this model, but not the only one. There are multiple reasons for building a groundwater model instead of using a single storage-outflow reservoir in land surface models: 1) Indeed, also in case of confined aquifers there is considerable interaction between surface water and groundwater. This is difficult to parameterize in a conventional land-surface model. 2) Our ultimate goal is also to estimate head declines. This is important because one needs to know how the depth-to-groundwater develops with time. If the groundwater head falls substantially ordinary farmers' pumps cannot reach it anymore and eventually it becomes unattainable for ordinary industrial pumps as well. It therefore constitutes a clear limit to groundwater availability. 3) Even though the volumes of groundwater traveling across catchment boundaries is limited, it becomes more and more important if one goes to even higher resolutions, and certainly in case of groundwater abstractions. 4) In many areas of the world groundwater significantly contributes to evaporation through groundwater convergence and capillary rise (see Fan et al., 2007; Bierkens and Van den Hurk, 2009) which warrants modeling groundwater head explicitly.

So in conclusion: this study adds considerably to previous depletion estimates and provides the first global estimates of global groundwater head variability and head decline and is a first step toward assessing how long our groundwater reserves could last.

Hereafter we answer other specific points raised by the reviewer:

Page 1. Line 1. This sentence appears to be the justification for building a global groundwater model, but this sentence is misleading. If by basins they mean surface water basins, then lateral flow is only significant in the most surficial part of the system

where >90% of the flow occurs. Most substantial groundwater withdrawals are from deeper confined and semi-confined aquifers where the total natural flow in the system is very small compared to the recharge, for example. It is true that most hydrological models do not include a groundwater flow component, but that in itself does not justify building a global groundwater model. Basically as the saying goes, "all models are wrong, but some are more useful than others"–so the authors have to demonstrate that their model is both accurate "enough" and "useful" to some degree. Unfortunately don't believe this model is either. It is only in the ballpark where the data exist in the 10% of the world where most of the pumping is occurring (it does a very bad job in much of the world in simply fitting depth to the water table), and it really isn't useful in the 10% of the world where the pumping is occurring because global groundwater depletion has already been quantified to a similar degree of accuracy (Wada et al, 2010, 2012; Konikow, 2011), and the groundwater model they are constructing really doesn't improve on those estimates. Lately it seems to be that the bigger the model is, the better it must be and the more likely we are to learn something new from it. Therefore the authors are now producing a global model; but in this case the model does not produce any useful results. More on this later.

We agree with the reviewer that, albeit flowpath analysis (See Figure 9 and also De Graaf et al., 2015) shows that many flowpaths passing catchment boundaries, the associated volumes are limited. However, we also observe a trend that future land surface models will operate at increasingly higher resolution (Bierkens et al., 2014), which makes that the across boundary fluxes at the grid scale will become more and more important, particular in case of groundwater abstractions. Our modeling effort is part of preparing for that circumstance. Also, as argued above (point 3) there are many other reasons for wanting to simulate groundwater heads at larger scales. We will more explicitly provide these reasons in our revised version

Page 1, line 19. This has been known for decades-why cite a reference from 2016? Why not a UN report from the 20th century?

The reviewer is right and we will add a reference to that effect.

Page 2, line 29. Here the problem begins. Indeed most abstractions occur in confined aquifers (which is why the investigators include a second layer to represent them), but simulating head declines there does NOT improve any estimates of groundwater depletion. Most all groundwater depletion in deep aquifers comes out of storage, not from bounding surface-water bodies, and therefore the model -simulated depletions only mimic the abstractions that are put into the model a priori. Thus the model is only as good as the estimate of water abstractions that are available. The earlier global depletion studies did their best to estimate these abstractions. Putting these numbers into a groundwater model does not improve on them.

As argued above under point 1), it is not true that most of the water comes out of storage. Most of the abstractions in our model indeed stem from confined aquifers or in confining layers, but they certainly attract groundwater at the expense of baseflow or river runoff as shown in Figure 12 and Table 3.

Figure 5 horizontal axes need labels (year). "Data" is not an acceptable y-axis labels aren't these heads, in meters?

The reviewer is right. We apologize for the oversight and will change this in the revised version.

Figure 6 is very hard to understand because the x-axes are not labeled and the y-axes are underlabeled. Count of what?

We will put the required labels in these figures in the revised version.

Figure 7. The points are too faint. It's not clear how far the simulated heads actually fall below the measured heads. Axis labels need to include "in meters".

The points have been deliberately plotted a bit faint to show the effects of point density in the Figure. These are thousands of points, so it is difficult to indentify individual points back. We will try to improve and put "(in m)" on the axes.

Figure 8. These results truly show how bad this model is. Only in the deserts of the world is the depth to the water table typically greater than 100 m. Yet most of the mountain ranges here show depths greater than 300 m. There are streams in most of those mountains so we know the depth to water is zero at those locations and likely a few tens of meters at the watershed divides. If so much of the world gives results this bad it does not give me confidence in the rest of the world, and there really isn't any reason to do a model of the whole world, especially when 90% of the abstraction occurs in 10% of the area. The authors will argue that these areas don't have any significant depletion anyway, but that argues that those areas really don't need to be modeled. If the time and resources were focused on the 10% of the world where depletion was really occurring, you would get better results and might learn something.

As argued under point 2 above, we clearly state that we are not able to simulate the perched water tables in hillslopes and the water tables in alluvial pockets in mountain valleys (we hope to resolve these in the near future), but calculate the groundwater in the bedrock below. The conclusion that as a result the results should be therefore bad in areas where groundwater tables are shallow is thus not at all founded, and we have the validation results to show that the model performs adequately in these areas. We do not think it is a good idea to limit the model to areas where there is depletion because: 1) we expect other areas that have yet no depletion to develop this in the future (parts of Africa, especially cities in deltas); 2) as stated under point 3 above, the groundwater model has other purposes than calculating depletion rates. Finally, in a previous paper (De Graaf et al., 2015) about the one-layer steady state version of the model we have masked the mountain areas out. We elected not to do this here, but could do this again.

Figure 11. This figure makes no sense. If you have an areal map then depletion volumes should be represented by meters, unless what they mean is cubic kilometers per square kilometer, but given that the numbers are up to 100 that can't be the case so this figure is undecipherable. In the paper on page 11 the top sentence I see now

the authors do indicate it is km3 per grid cell, but their grid cells vary in area across the globe so plotting results in this way is not consistent, nor are the number intuitive. M3/m2 would be much more meaningful.

This is a valid point: we will change this to M3/m2 in a revised version.

Table 1. The authors report using Sy values of 0.01 and higher for hard rocks, although in the text they way a storage coefficient of 0.001 was used for confined aquifer. Given the storage coefficient is the specific storage (which can vary by a couple of orders of magnitude) times the thickness (also quite variable) it is easy to imagine that the results could be off by a factor of ten. I realize the authors test the sensitivity by varying it by several factors, but given the authors are unable to quantify this uncertainty, it makes their results equally uncertain. Coming up with a "best-fit" scenario does not indicate their parameters are at appropriate values.

We calibrated the model to fit the head time series and we show the uncertainty in Figure 4. We will discuss the uncertainty more extensively in the discussion. We will refer to De Graaf et al (2015) where we already investigated the coefficient of variation of results (a measure of uncertainty) as a result of both conductivity and aquifer thickness uncertainty.
* * *